behaviour, evolution

motion dazzle, evolution, motion perception, citizen science, genetic algorithms

**Authors for correspondence:**
Anna E. Hughes
e-mail: anna.hughes@essex.ac.uk
Laura A. Kelley
e-mail: l.a.kelley@exeter.ac.uk

†Previous address: Department of Physiology, Development and Neuroscience, University of Cambridge CB2 3EB, UK.
‡Previous address: Department of Psychology, University of Cambridge, Cambridge CB2 3EB, UK.

# The evolution of patterning during movement in a large-scale citizen science game

Anna E. Hughes[1,†], David Griffiths[3], Jolyon Troscianko[2] and Laura A. Kelley[2,‡]

[1]Department of Psychology, University of Essex, Wivenhoe House, Colchester CO4 3SQ, UK
[2]Centre for Life and Environmental Sciences, University of Exeter, Penryn Campus, Penryn TR10 9FE, UK
[3]FoAM Kernow, Penryn TR10 8AE, UK

AEH, 0000-0003-2677-1965; LAK, 0000-0003-0700-1471

The motion dazzle hypothesis posits that high contrast geometric patterns can cause difficulties in tracking a moving target and has been argued to explain the patterning of animals such as zebras. Research to date has only tested a small number of patterns, offering equivocal support for the hypothesis. Here, we take a genetic programming approach to allow patterns to evolve based on their fitness (time taken to capture) and thus find the optimal strategy for providing protection when moving. Our 'Dazzle Bug' citizen science game tested over 1.5 million targets in a touch screen game at a popular visitor attraction. Surprisingly, we found that targets lost pattern elements during evolution and became closely background matching. Modelling results suggested that targets with lower motion energy were harder to catch. Our results indicate that low contrast, featureless targets offer the greatest protection against capture when in motion, challenging the motion dazzle hypothesis.

## 1. Introduction

The high contrast, conspicuous patterns seen on animals such as zebras have attracted a range of evolutionary explanations, including camouflage, thermoregulation, communication, and the avoidance of biting flies [1–7]. One hypothesis that has received attention in recent years is the 'motion dazzle' hypothesis, which proposes that these patterns may act to cause confusion when the animal is in motion, causing illusions in the visual system of the viewer that may lead to misjudgements of speed and direction [8].

There have been a number of studies that have provided support for the motion dazzle hypothesis. For example, it has been shown that putative dazzle patterns are relatively difficult for humans to 'catch' in a computer-based touch screen game [9–11], and may also interfere with speed [12–14] and direction [15] perception. There is also evidence that some orientations of stripes can interfere with the ability to track one target within a larger group [16–18]. Finally, modelling work has suggested that striped patterns may be particularly prone to creating erroneous motion signals in the visual system, which may underlie these types of behavioural effects [19].

Despite these findings, not all research has supported the motion dazzle hypothesis. Some studies on humans have found that striped targets are easier to capture than non-patterned targets [20,21], and moving cuttlefish have been shown to preferentially display low contrast patterns [22]. Similarly, a recent study using natural predators hunting patterned prey found no evidence for a benefit of motion dazzle patterning compared to uniform colouration [23]. Some studies that have argued for an effect of motion dazzle patterning have shown that there is no benefit in terms of capture success of striped patterning

**Figure 1.** Figure showing screenshots from the game. Left: title screen. Middle: instructions presented to the participant. Right: the game in progress. Participants could see the time remaining on the trial via the countdown clock in the top left-hand corner.

over a luminance matched non-patterned target, suggesting that the benefit of stripes may not be unique [9,11,12,21].

One potential limitation of previous studies is that they have tested a relatively small range of patterns, often chosen arbitrarily. This means that it is not yet clear whether we have truly discovered the optimal patterning type to provide protection when in motion; it may be that there are more effective options than those tested so far. However, the small-scale psychophysics-style experiments used to date make it difficult to test large numbers of patterns. Even for striped targets, included in all previous studies on motion dazzle, only a limited range of spatial frequencies, contrasts, and orientations have been tested. We, therefore, took a novel approach, using genetic programming to allow the patterning of targets to 'evolve' across generations in response to capture success [24–26]. In this way, we can ask which patterning strategy is optimal, given the almost infinite number of possible patterns that can be generated. To obtain the large amount of data required for this approach, we ran our experiment as a citizen science game (Dazzle Bug) in a popular visitor attraction. Participants played the game by tapping on the moving targets (bugs) with their finger as quickly as possible in order to 'catch' them (figure 1). We ran a number of replicates of the evolutionary process for three populations of different speeds, to assess whether the optimal patterning changes as a function of the target movement speed.

Our first aim was to demonstrate a fitness increase in our experimental populations, which we defined as an increase in the average capture time across generations. We did this by comparing to a simulation run of the evolutionary algorithm, using randomized capture times. We then investigated how the target patterning changed across generations for different speed populations, using image analysis to measure contrast and the presence of stripes at different orientations. We also looked at whether selection rates differed for the different speed populations, using the Lande, Arnold, and Wade framework [27–29], allowing us to consider how selection pressure might vary across the generations. Finally, we asked whether motion perception modelling can help to explain our experimental results.

## 2. Methods

### (a) Subjects
We did not collect any demographic data from participants. This was to streamline participation in the study (which was conducted in a busy exhibition space) and also because it would be difficult to verify the accuracy of the information presented. To overcome the limitations of being unable to account for participant age, handedness, and gender, we collected a large sample size of participants over many generations (1 554 935

targets were caught in total across the whole experiment, involving approx. 75 000 participants).

### (b) Experimental methods
The Dazzle Bug game was installed at the Eden Project (St. Austell, UK) on a touch screen computer as part of an interactive exhibition, and the data used were collected between May 2018 and January 2019. The game was coded in HTML5 canvas (source code and images are available at https://github.com/foam/dazzlebug/, DOI: 10.5281/zenodo.2560935) and is playable online at dazzle-bug.co.uk/exhib.html (the online data are not analysed in this paper). The screen had an area of 478 × 269 mm and the screen resolution of the game was 1237 × 820 pixels. The viewing distance of participants to the screen was approximately 60 cm (based on observing visitors playing the game). However, we did not attempt to control this strictly because of the nature of the event space, and because viewing distances would not be standardized in a more naturalistic situation.

The game had a similar format to many previous studies testing motion dazzle effects [10,11,21] in that participants were presented with a small rectangular target (75 × 100 pixels, or 29.0 × 38.6 mm; visual angle 2.8 × 3.7°) which they had to try to 'catch' as quickly as possible after it had appeared by touching it with their finger (figure 1). Targets began their movement at a random position on the screen and moved immediately upon presentation with a linear trajectory. The angle of movement changed throughout a trial, both at the edge of the target arena via reflection (to ensure that the target remained visible to the participant) and randomly throughout the movement (once every half a second, and when an unsuccessful capture attempt was made; the new angle was randomly chosen to be within 90° of its previous angle). Targets could be presented at one of three speeds, fast, medium, or slow (600, 450, or 300 pixels per second, respectively, which equated to 231.8, 173.8, and 115.9 mm s$^{-1}$ or 23.4, 17.6, and 11.7 deg s$^{-1}$). These speeds were similar to those used in previous studies [11,13–15,20,21]. Each participant was presented with a random mix of targets of all three speeds. Participants had 5 s to catch each target. After the target had been caught, or the time-out limit had been reached, the game would move automatically onto the next target. A game consisted of 20 trials in total, with the targets presented randomly selected from the current generation.

#### (i) Background photos
Targets were presented against one of 40 naturalistic background photographs (e.g. grass, tree bark, or leaf litter). The background was randomly selected on each trial (previous work using similar images has shown that the effect of background type on capture rates is small [21]). The photos were converted to greyscale (with an average pixel value of 127).

#### (ii) Pattern generation
The patterns throughout the game were generated through a genetic programming approach [24–26]. This does not attempt to directly mimic biological evolution but is instead a method

allowing the exploration of an unbounded parameter space in an efficient manner, using algorithms inspired by natural selection processes. The key principle is that the evolutionary process acts to modify small 'computer programs' that specify the patterning presented on each target, thus generating targets with patterns based on complex manipulations of a set of starting images. This allows a great deal of flexibility in the complexity of target patterning and reduces artificial bounds on the evolutionary space that can be introduced in more traditional genetic algorithm methods [25]. For full details of the pattern generation process, see the electronic supplementary material (S1).

### (iii) Evolutionary process

Four replicates of the game were run, with each replicate containing three separate populations for each speed (fast, medium, and slow) that each evolved separately. The first generation of each population contained 128 individuals that were completely randomly generated in accordance with the pattern generation process detailed above. These were then presented to players randomly until they had all been played five times. At this point, each one was scored by averaging the time taken to catch them, and this was used as a measure of fitness (i.e. targets that took longer to catch had higher fitness). Normalization of participant times or removing censored data (i.e. trials on which the target had not been caught) was not possible due to the design of the evolutionary algorithm. The bottom half of the generation based on this measure of fitness was removed from the population. The top 64 targets were copied with no mutation to form one half of the new generation, and then copied again with a small probability of mutation to form the other half (see electronic supplementary material, S2 for full details of the mutation process). This type of 'overlapping replacement' is thought to perform best, and allows the best bugs from each generation to be tested against newly introduced individuals [30].

The exact number of generations tested varied between replicates because each participant was randomly assigned to one replicate, and because not all replicates were run simultaneously. Replicate 1 had 89 generations, replicate 2 had 87 generations, replicate 3 had 45 generations, and replicate 4 had 46 generations (each replicate was started and ended manually, leading to different numbers of generations.)

### (iv) Control model

We ran a control model to confirm that any systematic patterning changes seen during the real game were due to directional selection, rather than drift or biases within the genetic programming algorithm. This was set-up identically to the real experiment, except that instead of participants playing the game, the computer randomly selected a 'capture time' for each target in each generation, based on a lognormal distribution using the mean and standard deviation of each population in the real experiment (as individual clicks were not recorded in our experimental data, we estimated the variance of each individual trial by multiplying the variance of the 'bug-level' fitness by the number of plays of each bug, e.g. by 5, reflecting the fact that the 'group-level' variance was calculated by dividing by the sample size). The null model was run for 40 generations.

### (c) Pattern quantification

We analysed the patterning of the targets using custom-written scripts in ImageJ (v. 1.51 k) [31]. This script first calculated the mean, minimum, and maximum luminance of each target, and the standard deviation of the luminance. We also calculated the contrast of the target as the coefficient of variance in luminance (the standard deviation divided by the mean). We then used convolutional Gabor filtering methods that allow the measurement of different angles at different spatial frequencies to determine the strength of these signals on the targets in a biologically plausible way [32–35]. We analysed four angles (vertical, horizontal, and

two diagonal stripes) each at four different spatial frequencies (sigma values of 2, 4, 8, and 16 pixels, or 5.6, 2.7, 1.4, and 0.7 cycles/deg). The spatial frequencies were chosen to align well with the peak of the human contrast sensitivity function [36–38]. Four angles were selected for the Gabor filter as this is the minimum number required to capture all relevant angle information [39]. For each of these conditions, we calculated the standard deviation of Gabor-convolved pixel values as a measure of the 'energy' at that particular angle and spatial frequency. Finally, we also measured the standard deviation of Gabor-convolved pixel values for a rectangle covering the edge (with a width equal to sigma) at an angle orthogonal to the edge for all four edges of the target (top, bottom, left, and right). This allowed us to investigate whether the placement of patterning has an effect on fitness; for example, it has been suggested that stripes on the leading edge of a target may redirect capture attempts posteriorly [13].

### (d) Statistical analysis

Data analysis was run in R (v. 4.0.3) [40] and linear mixed models were fitted using lme4 (v. 1.1–25) [41]. We used only data from bugs that were attempted at least five times, and we also removed any impossible reaction times (e.g. those above the time-out threshold; these were rare, and including them in the analysis did not alter any of the main conclusions). We expected many of the patterning measures to be autocorrelated and therefore we reduced the number of variables by determining which were the best predictors of capture time using a model selection approach via linear mixed modelling. This gave us a total of five pattern metrics (one each for luminance, vertical stripes, diagonal stripes, and an 'edge' metric).

Based on residual plots, we used the natural logarithm of fitness as our dependent variable in our statistical models. First, we generated a linear model of fitness across generations (with population, i.e. target speed as a fixed effect and replicate as a random intercept) to test whether there was a change in fitness across generations. For this model, we used all the experimental data.

For the remaining analyses, we used the first 40 experimental generations in each replicate only. We compared the change in fitness of our targets across generations for both the Eden project data and the null data, allowing us to test whether fitness improved in our experimental population compared to a null baseline. This was done by extending the previous model to include a variable which coded whether the data point belonged to an experimental or a control population. We next tested whether there were differences in how our five patterning measures had changed in the experimental and the null populations using cumulative link models with generation as an ordinal dependent variable and the patterning measure, the 'control/experimental' variable, and the interaction between them as the independent variables.

Finally, we analysed whether there were any differences in selection rates for the different speed populations in the experimental population. To do this, we used the Lande, Arnold, and Wade framework [27–29] to calculate linear selection rates ($\beta$) for each of the five patterning measures within each population. $\beta$ gives a measure of selection pressure across the generations, allowing us to determine which phenotypic characteristics were experiencing strong selection pressure and when. We used these to test for differences in linear selection rates between different speed populations and over evolutionary time (generations). Full details of all statistical analysis (including tables of unstandardized effect sizes for all models) are available in the electronic supplementary material (S3).

### (e) Motion modelling methods

Motion modelling was carried out using a MATLAB implementation of a motion model using a two-dimensional array of correlation-type elementary motion detectors (as described in [42] and available at https://github.com/AdamPallus/2dmd) [19,43,44]. For each 'fast' bug in generation 0 (512 bugs in total)

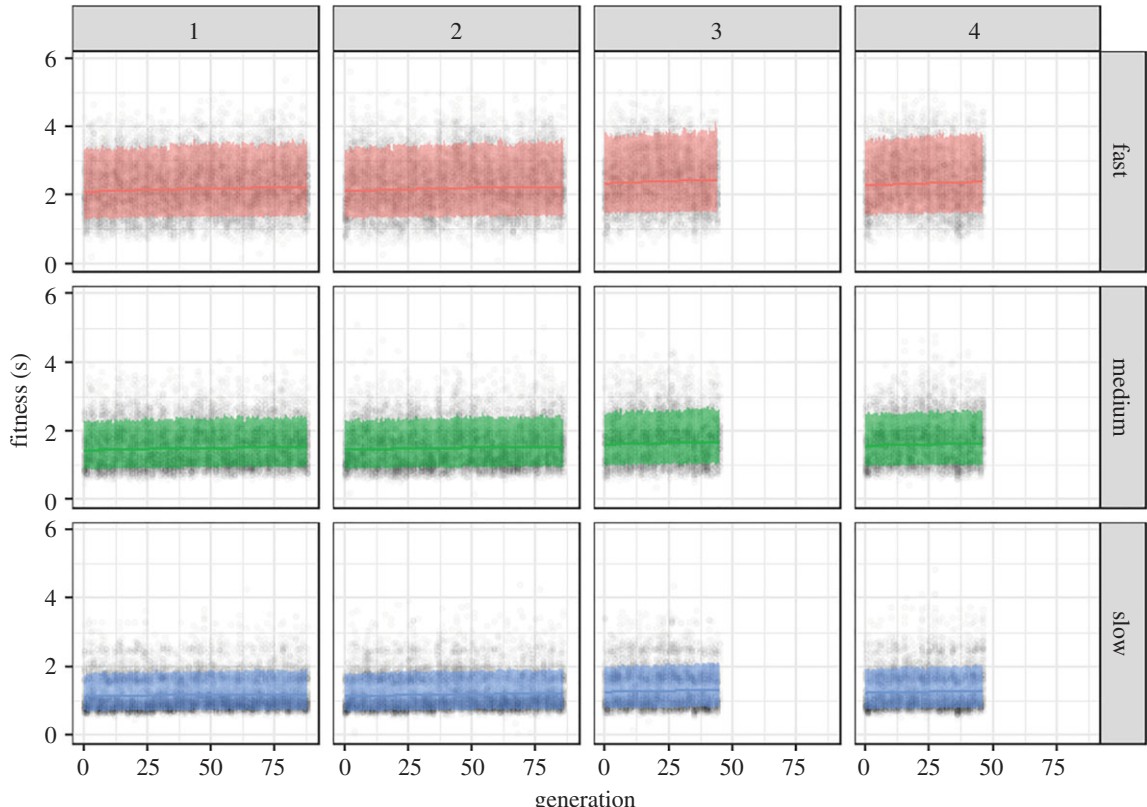

**Figure 2.** Experimental data from all four replicates and all three speed populations, showing how fitness (i.e. average capture time) varies with generation number. Raw data points are included as dark circles. The lines are fitted as quadratic polynomials on the predicted data from the model, and the confidence bands are 80% prediction intervals. (Online version in colour.)

we generated a short movie where the bug initially moved on an upwards trajectory and then rotated to move on a trajectory 15° to the right (see electronic supplementary material for an example). We used the generation 0 bugs as these should display a wide range of randomly selected pattern types, and the 'fast' population as selection seemed to be strongest on these targets, suggesting that we should see the largest differences in fitness for this population. Details of the model parameters and set-up are provided in the electronic supplementary material (S4).

The model outputs were a measure of motion direction (theta) and magnitudes (radius) for each pixel in each frame of each video. For each video, several circular statistics metrics were calculated from these outputs (after removing zeros, corresponding to places in the image where no motion signal was observed). Firstly, the mean resultant length of the circular direction data (theta) was calculated. This is a statistic between 0 and 1 that gives information of the spread of a circular variable, and thus gives a measure of motion coherence [45]. Secondly, the average vector length was calculated as a measure of motion energy. Finally, the bias was calculated by taking the difference between the circular mean (i.e. the average direction) and the 'veridical' trajectory of the target (assumed to be the average of the two directions the target moved in during the trial). All circular statistics were calculated using CircStat [46]. Details of the statistical analyses are provided in the electronic supplementary material (S4).

## 3. Results

### (a) Is there a fitness increase for the experimental populations, and does this differ from the null population?

Figure 2 shows there were clearly large differences in fitness (capture speed) among populations, with the fast bugs being

hardest to catch, followed by the medium bugs, and then finally the slow bugs ($\chi^2 = 50892.85$, $p < 0.001$). There was a considerable level of noise in the data, which is to be expected given the wide range of participants and fast reactions required. Nevertheless, there was also a significant increase in fitness across generations ($\chi^2 = 208.72$, $p < 0.001$). Increases were often particularly obvious in the early generations of the game. The experimental data also show a significant difference in fitness change compared to the null data (interaction between dataset and second-order effect of generation: $\chi^2 = 118.959$, $p < 0.001$). The experimental data shows an initial increase that flattens off (electronic supplementary material, figure S4); using predictions from the model, there is an approximately 170 ms increase in fitness in the experimental fast population in the first 10 generations, compared to a 30 ms increase in the control fast population. We, therefore, have evidence for a small fitness increase in our experimental population, suggesting that selection is occurring to optimize patterning types.

### (b) How does bug patterning change in the experimental and null populations?

All four populations of evolving bugs demonstrated a loss of pattern information over the generations—converging on uniform background-matching colours (figure 3, top)—while the control populations maintained their pattern diversity (figure 3, bottom). Quantifying this using our five most informative pattern measures (figure 4) shows that there are always clear differences between how the pattern measures change in the experimental condition compared to the control condition (interaction between experimental/control condition and

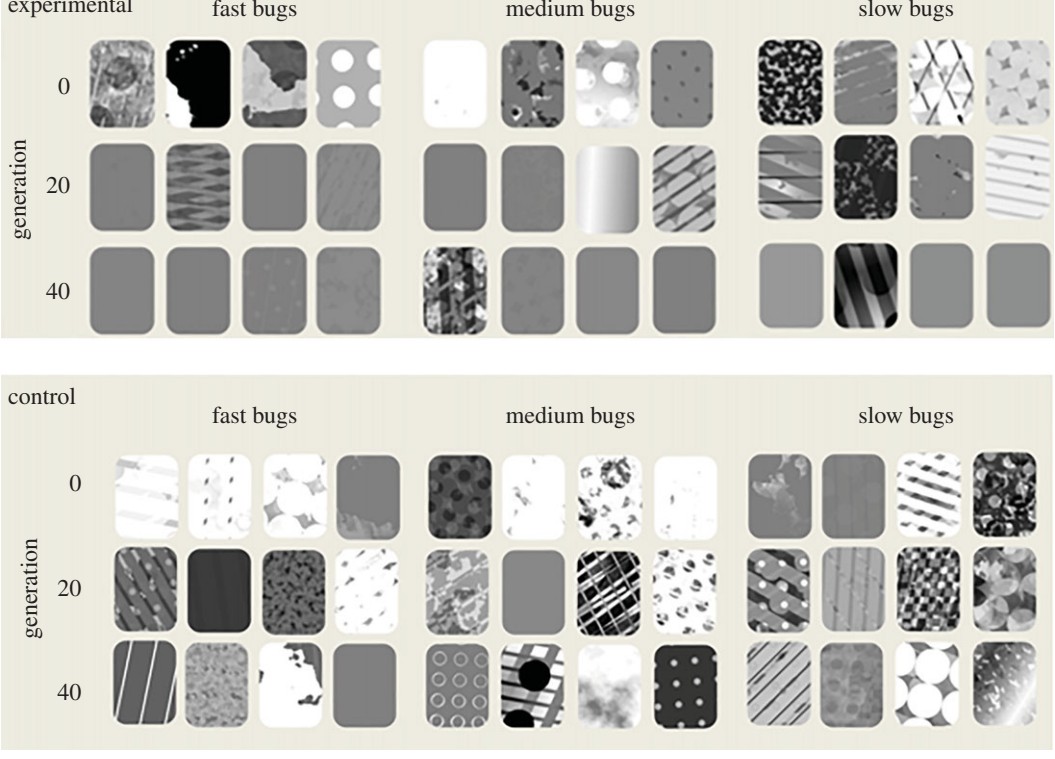

**Figure 3.** Top—random bugs from generations 0, 20, and 40 (all from the same replicate) of the experimental data, split into populations (fast, medium, and slow). Bottom—random bugs from generations 0, 20, and 40 (all from the same replicate) of the control data, split into populations (fast, medium, and slow).

pattern measure for cumulative link models—standard deviation of bug luminance: $\chi^2 = 36283.7$, $p < 0.001$; vertical stripes: $\chi^2 = 36460.1$, $p < 0.001$; horizontal stripes: $\chi^2 = 36909.5$, $p < 0.001$; diagonal stripes: $\chi^2 = 36395.6$, $p < 0.001$; right edge: $\chi^2 = 36857.6$, $p < 0.001$). Broadly, there always seems to be an overall decrease in pattern complexity in the experimental case, whereas there is much more variability in the control condition.

### (c) Are there differences in selection rate for each speed population?

The data allow us to determine the main selection pressures operating on each population of bugs within each generation (normalized linear selection rates ($\beta$)), so that we can assess whether pressures change over evolutionary time. Differences in selection rates across generations (i.e. the polynomial term for generation was significant) were seen for luminance ($F = 6.336$, $p = 0.002$), vertical stripes ($\chi^2 = 13.516$, $p = 0.001$), and for diagonal stripes ($F = 3.472$, $p = 0.032$). There was no evidence for difference in selection rates for both the horizontal stripe ($\chi^2 = 1.628$, $p = 0.443$) and the right edge measures ($\chi^2 = 5.703$, $p = 0.058$).

The standard deviation of the luminance of the bugs appears to be particularly important for the 'fast' population; there is strong selection pressure particularly in early generations, and this differs from the selection rate seen in the 'medium' and 'slow' populations (figure 5; fast-medium comparison: $t = -3.189$, $p = 0.004$; fast-slow comparison: $t = -3.504$, $p = 0.001$; medium-slow comparison: $t = -0.315$, $p = 0.947$). For vertical stripes, there is some evidence for stronger selection pressure for medium compared to slow bugs ($t = -2.482$, $p = 0.036$), and for horizontal stripes, there is evidence for stronger selection pressure for fast compared to slow bugs ($t = -2.456$, $p = 0.038$). For all other patterning

parameters, there were no significant differences between the different speed populations (diagonal stripes—$F = 0.954$, $p = 0.386$, right edge—$\chi^2 = 5.687$, $p = 0.058$).

### (d) Can motion modelling help to explain the experimental findings?

According to previous modelling work [19], we would expect targets to produce strong motion illusions if they are both highly coherent (the motion vectors produced tend to be in a highly similar direction) and biased (the average trajectory of the motion vectors is quite different from the 'veridical' direction of the target). In our model, we found a significant interaction between coherence and bias in predicting the fitness of targets ($F = 5.985$, $p = 0.015$). When visualizing the most coherent targets there appears to be an increase in fitness as the bias increases, in line with previous predictions [19]. In addition, the targets with the highest bias also tended to be relatively stripy and high contrast (bugs with higher bias had both higher standard deviations of luminance $F = 10.844$, $p = 0.001$, and levels of vertical stripes $F = 35.688$, $p < 0.001$) again suggesting that these 'motion dazzle' type patterns might be expected to create illusory motion signals.

However, these results do not seem to explain our evolutionary findings, where we saw a strong tendency for targets to become lower contrast and non-patterned. A second metric from our motion modelling is the motion energy, which can be conceptualized as how salient or visible the motion is. Here, there is a very different relationship with fitness, as can be seen in figure 6 (bottom), with low motion energy targets (that tend to be low contrast and have little patterning) having higher fitness than those with higher motion energy (that tend to have high contrast and strong patterning) ($F = 4.939$, $p = 0.027$; $F = 4.988$, $p = 0.026$ if data were not filtered to exclude cases with a circular mean difference of greater than

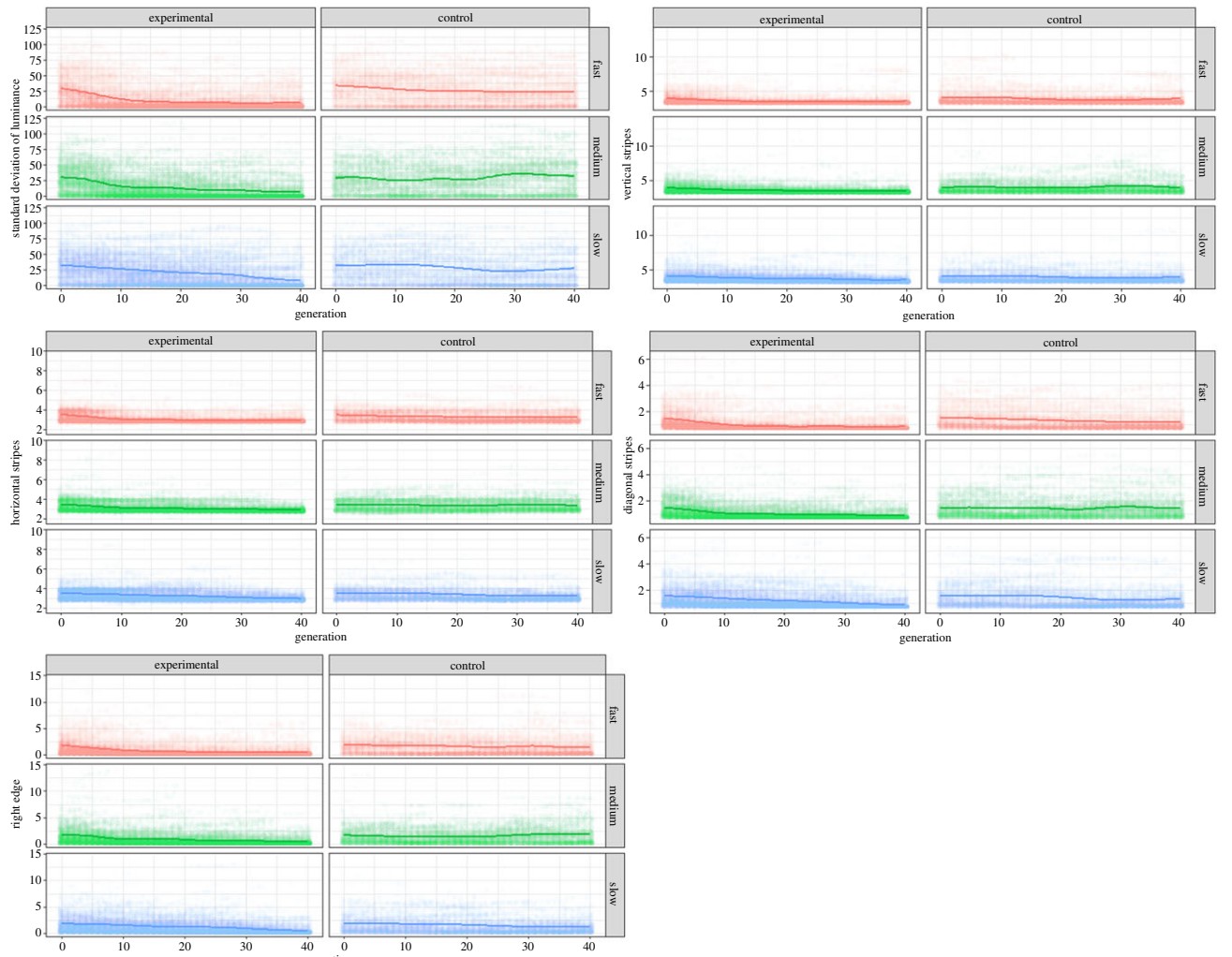

**Figure 4.** Change in selected parameters across generations (chosen via model selection), both for the experimental (left) and control (right) conditions across all replicates, for five pattern measures: top left—standard deviation of the bug luminance, centre—the pattern energy for the vertical stripes, bottom right—horizontal stripes, bottom left—diagonal stripes, and top right—the right edge. The lines represent generalized additive model (GAM) fits and are provided to give a visual cue to the general trend in the data (they do not represent the model fits). (Online version in colour.)

6°). Bugs with higher mean vector lengths had both higher standard deviations of luminance ($F = 1171.8$, $p < 0.001$) and levels of vertical striping ($F = 545$, $p < 0.001$).

## 4. Discussion

Using a large-scale evolutionary citizen science game, we found no evidence that putative 'motion dazzle' patterning can offer protection when in motion. Despite predictions that high contrast, geometric patterning should cause visual illusions that make targets harder to catch, we found that the targets consistently evolved to become less patterned and lower contrast. This happened for all speeds tested and all replicates of the experiment, although these changes seemed to occur more quickly in populations with faster speeds. While the increase in fitness was small in absolute terms, it reflected an approximately 10-fold increase in capture time compared to a null model. In addition, fast reactions are often seen in the context of predator–prey reactions (e.g. cuttlefish prey seizure time is approx. 60 ms [47], and praying mantis show escape responses to bat predators in approx. 114 ms [48]), highlighting that our findings are biologically plausible. Motion modelling suggested that these results

could be a consequence of the motion energy of the stimulus, as this was correlated with capture time, with lower motion energy targets being more difficult to catch. Our results have important consequences for our understanding of the evolution of stripes, and for how animals should best protect themselves from capture when in motion.

Our results are perhaps surprising in the context of most literature on motion dazzle to date, which has suggested that stripes seem to be relatively difficult to catch or can cause illusions of speed or direction perception [9–12,14–18]. However, we note that there has indeed been plenty of evidence in the literature for uniform grey patterns also being relatively difficult to catch, and in some cases perhaps even harder than striped targets. For example, grey targets seem to survive similarly or better than striped targets in capture studies [9,11,12,21]. Similarly, in tracking tasks, low contrast parallel stripes were found to be more difficult to track than high contrast parallel stripes [18], arguing against a motion dazzle explanation. Recent work has also suggested that in some cases striped patterns are only difficult to catch when the targets are moving sufficiently quickly to blend via the 'flicker-fusion' effect into uniform grey [49]. An average striped target in our game would have had a temporal frequency of approximately 30 Hz, below the range of human

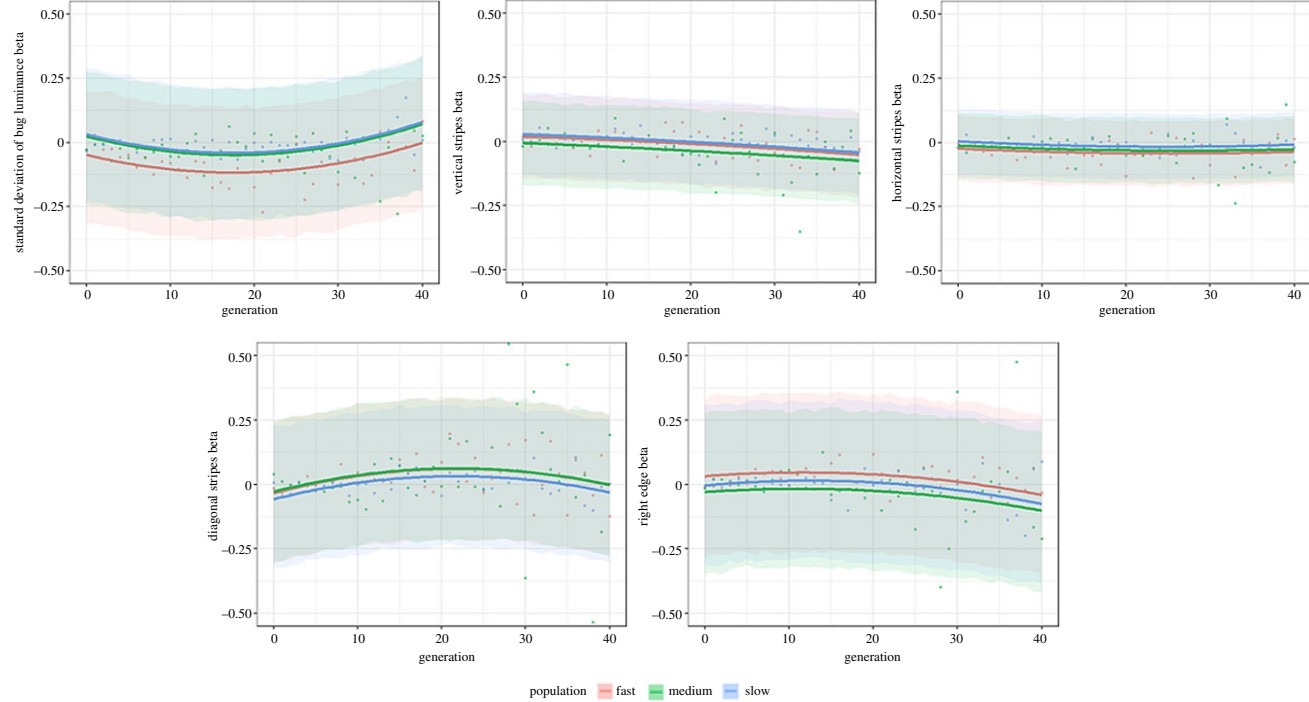

**Figure 5.** Normalized linear selection rates ($\beta$) for the five most important patterning measures across generations. Individual points show the selection rates, averaged across each replicate. The lines are fitted as quadratic polynomials on the predicted data from the model. Confidence bands are 80% prediction intervals. (Online version in colour.)

achromatic flicker-fusion thresholds, which are thought to be between 35 and 60 Hz [50,51], suggesting they would not have benefited from a flicker-fusion effect. Our results, therefore, suggest that uniform grey targets had a survival advantage over other types of target patterning, leading them to become fixed as the optimal strategy in all our populations, regardless of speed or replicate number.

Motion modelling has previously suggested that stripes should create erroneous motion signals that are both highly coherent and biased [19], implying striped prey should be more difficult to catch. However, to our knowledge, modelling results have not previously been compared to behavioural data. Our large dataset, therefore, offers a perfect opportunity to study whether the motion modelling results do indeed correlate with capture times. In support of the motion dazzle hypothesis [19], we do indeed find that highly coherent and biased targets tend to be more difficult to catch than less biased coherent targets, and that the most biased and coherent targets are often stripy. However, this clearly does not explain the results we see in the evolutionary game. We thus considered another metric that can be calculated from motion models, namely the motion energy, and found that this also correlated with capture success. Targets with low motion energy (that tended to be uniform grey) were harder to catch than targets with high motion energy (that were much more high contrast and patterned).

Why does reducing motion energy seem to be a better predictor of the outcomes in our evolutionary games compared to motion dazzle strategies which maximize the bias/coherence metric? We speculate that motion energy is a very consistent signal; regardless of the trajectory of the bug or the speed, the targets with low visibility will be harder to catch than those that are highly visible. This is likely due to both increased difficulty in tracking, and also increased difficulty in detecting the target, and may reflect the fact that

these targets are often low contrast and may 'blur' into the background. We suggest that stripes may reduce tracking ability more than detection and thus any effects may be much more dependent on the particular orientation of the stripes, given that the most effective striped targets appeared to have relatively similar dominant orientations (figure 6, top), and previous studies have shown orientation dependence for the effects of striped targets [16–18,21]. Small mutations affecting the rotation of striped patterns could, therefore, potentially cause large changes in fitness, potentially making striped patterns a relatively unstable evolutionary strategy compared to uniform grey in our experiment. For this reason, the random movement of the targets in our experiment may have further reduced the fitness of striped patterning. Given that real striped animals, such as zebras, are unable to change course so rapidly, it would be interesting for future experiments to explore whether targets moving with a more consistent trajectory are more likely to evolve striped patterns.

We used three different speed populations in order to assess whether there were differences in the patterns that evolved. As expected, we found that there were strong differences in capture difficulty for different speed populations, with fast targets being the hardest to capture, but we did not find evidence for there being differences in the target patterning that evolved, with all populations becoming uniform grey. This is in agreement with previous work suggesting that there is no interaction between target speed and prey patterning [11], at least for speeds below that needed to create a 'flicker-fusion' effect. However, we did find increased selection in 'fast' populations, particularly early on in the evolutionary process for the contrast measure and later on for the vertical stripe measure. This may simply reflect the higher difficulty of these targets, which is likely to give a wider range of capture times and thus offer more variation for selection to operate on, potentially exaggerating the selection process.

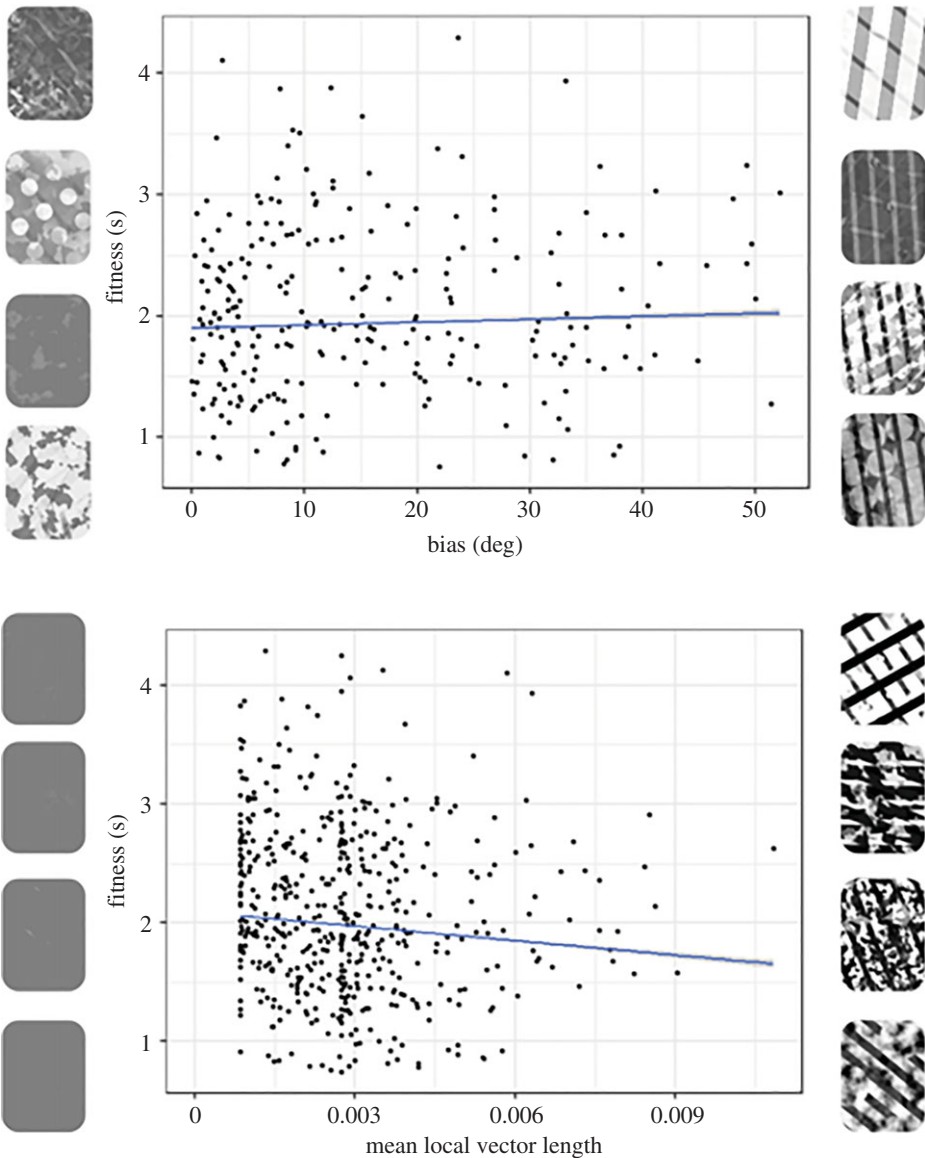

**Figure 6.** Top—data from the generation 0 fast bugs, using only those bugs above the median coherence value (i.e. relatively highly coherent targets) and plotting fitness against the bias. The bias was calculated as the difference between the average direction of the target as measured by the model and the 'veridical' motion direction. Exemplars on the left are bugs that had low bias values, according to the motion model; exemplars on the right are bugs that had high bias values. The confidence band is the 80% prediction interval for the predictions from the linear model. Bottom—data from the generation 0 fast bugs, plotting fitness against the mean local vector length (a measure of motion energy, or how salient or visible the motion is). Exemplars on the left are bugs that had low motion energy values, according to the motion model; exemplars on the right are bugs that had high motion energy values. The confidence band is the 80% prediction interval for the predictions from the linear model. (Online version in colour.)

Genetic algorithms are complex and there are many different ways to implement them [24–26]. We, therefore, carried out control experiments using simulated reaction time data with similar average distributions to the real data, helping us to rule out explanations of our results based on algorithmic biases or genetic drift. Our results show clearly that selection pressures do indeed operate in our game and that the change towards grey targets does not simply reflect drift. However, while our set-up allowed us to explore a very wide range of pattern types, it is possible that different algorithms could produce different targets and thus perhaps different results. For example, our targets were rarely highly asymmetric (although this was possible). Recent research has suggested that stripes may be particularly effective at misdirecting capture attempts when they are placed on the anterior of a target [13], suggesting that an interesting direction for future work could be to allow the algorithm to specify

different genes (and thus different patterning) for different parts of the target. Further speed manipulations could also be of interest; in particular, it would be instructive to compare the patterns that evolve on moving targets to those that evolve on stationary ones [10,52]. It is also possible that other aspects of the visual environment that we did not test in the current experiment could influence the relative efficacy of different strategies, such as the level of target occlusion. Finally, future studies could test the possibility of stripes serving multiple functions (such as distance-dependent protective colouration for aposematism and camouflage [53]).

Our experiment used human participants, in line with the majority of studies in this area. Of course, in the natural world, the viewing animals might have very different visual systems to humans, and the viewing conditions would also likely be quite different from the current experimental set-up. We removed colour cues from our experiment, as it is

well known that different species have very different colour perception [54,55], although motion vision is generally thought to be predominantly achromatic [56–58]. However, there is also large variability in the perception of temporal changes across different species [59] which we could not adequately compensate for in this experimental set-up. Despite this, our main conclusions broadly agree with previous studies carried out on non-human predators and prey [22,23]. However, it would of course be highly instructive to carry out similar experiments with non-human animal participants to determine whether the results we report here are more widely generalizable.

Overall, we find limited evidence for motion dazzle effects in a citizen science evolutionary game, which we believe is the most comprehensive test of this hypothesis to date. Stripes were able to cause motion illusions and reduce capture times in some scenarios, meaning that there may still be specific cases where motion dazzle can be at least part of an explanation for the evolution of striped patterns. However, our results suggest that uniform grey targets appear to be a more stable optimal solution.

Data accessibility. The code for the Dazzle Bug game is available at https://github.com/fo-am/dazzlebug/, doi:10.5281/zenodo.2560935. Spreadsheets of the full results used in analysis and analysis scripts are available at https://osf.io/s5wxy/. Images and videos used in this study are available from the corresponding authors on reasonable request.

Authors' contributions. A.E.H., D.G., and L.A.K. conceptualized the study, D.G. created the Dazzle Bug game, A.E.H. and J.T. analysed the data, and A.E.H. wrote the first draft of the manuscript. All authors read and approved the final manuscript.

Competing interests. We declare we have no competing interests.

Funding. A.E.H. was supported by a PhD studentship from the BBSRC (BB/F016581/1). L.A.K. received funding from the People Programme (Marie Curie Actions) of the European Union's Seventh Framework Programme (FP7/2007-2013) under REA grant agreement n° PIIF-GA-2012-327423 and is currently funded by a Royal Society Dorothy Hodgkin Fellowship (DH160082). J.T. is funded by a NERC Independent Research Fellowship (NE/P018084/1). The funding bodies had no role in the design of the study, the collection, analysis, and interpretation of data or in writing the manuscript.

Acknowledgements. We would like to thank Amber Griffiths for providing valuable assistance and the Eden Project for hosting Dazzle Bug. We are grateful to two anonymous referees for comments that greatly improved the manuscript.

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
