## [Reviewer comments · Proceedings of the Royal Society B: Biological Sciences]

Review History

RSPB-2020-0899.R0 (Original submission)

Review form: Reviewer 1

Recommendation

Major revision is needed (please make suggestions in comments)

Scientific importance: Is the manuscript an original and important contribution to its field?

Good

General interest: Is the paper of sufficient general interest?

Good

Quality of the paper: Is the overall quality of the paper suitable?

Excellent

Is the length of the paper justified?

Yes

Should the paper be seen by a specialist statistical reviewer?

No

Do you have any concerns about statistical analyses in this paper? If so, please specify them explicitly in your report.

No

It is a condition of publication that authors make their supporting data, code and materials available - either as supplementary material or hosted in an external repository. Please rate, if applicable, the supporting data on the following criteria.

Is it accessible?

Yes

Is it clear?

Yes

Is it adequate?

Yes

Do you have any ethical concerns with this paper?

No

Comments to the Author

This is a well-written, concise article with a very precisely documented methodology. I find the citizen science approach for data collection a versatile method and providing an off-the-shelf touchscreen software for other behavioural ecologists to run camouflage experiments would be of great interest. However, I have a few concerns regarding the paper which are listed below:

1. The authors should provide a readme file on the github repository that provides the necessary information on how to run the software locally. Right now, the readme only says "A new camouflage hunting evolution experiment where we incorporate movement. More info soon!". If the authors only wish to update this information upon publication (makes sense to me), it would be still useful to make the readme available to reviewers.
2. Between lines 369-377 the authors argue that the orientation of stripes could have a significant effect on fitness and small mutations can render stripes a relatively unstable strategy. I wonder whether the random movement of targets could play a significant role in the instability of stripes and their lower fitness compared to uniform grey. Classic examples of stripy targets (where stripes thought to give protection from attacks) like zebras and ships cannot change course so rapidly as the targets in this study and the orientation of their stripes remain consistent to their trajectory, i.e. neither zebras or ships can "fly" upwards/downwards from the predators point of view. It would be interesting to see if a more consistent trajectory (in terms of less angular changes) yields better fitness for stripy patterns. I am not necessarily suggesting further experiments, but I think at least the difference between randomly moving targets and beasts bounded by the laws of physics should be expanded more in the discussion. I guess a real-world example of a stripy animal that can very quickly change its trajectory would be a hoverfly, but as far as I remember there is an argument that they mimic bees...
3. I wonder if there is a correlation between the number of capture attempts / location of capture attempts and the patterns. Could the edges of relatively fast-moving uniform grey targets blur with the backgrounds while the edges of high contrast targets remain easily trackable? I remember that classic high contrast dazzle patterns should make the predator under or overshoot the target along its expected trajectory, however this effect could be significantly lowered by targets randomly changing trajectory while the "blurring" mentioned above might be independent from how fast directional changes occur (if they occur at all).

Review form: Reviewer 2 (Benedict Hogan)

Recommendation

Accept with minor revision (please list in comments)

Scientific importance: Is the manuscript an original and important contribution to its field?

Excellent

General interest: Is the paper of sufficient general interest?

Good

Quality of the paper: Is the overall quality of the paper suitable?

Marginal

Is the length of the paper justified?

Yes

Should the paper be seen by a specialist statistical reviewer?

No

Do you have any concerns about statistical analyses in this paper? If so, please specify them explicitly in your report.

Yes

It is a condition of publication that authors make their supporting data, code and materials available - either as supplementary material or hosted in an external repository. Please rate, if applicable, the supporting data on the following criteria.

Is it accessible?

Yes

Is it clear?

No

Is it adequate?

Yes

Do you have any ethical concerns with this paper?

No

Comments to the Author

This is an ambitious and well designed experiment that uses an evolutionary algorithm to find the best pattern for escaping capture in a human-predator style experiment. I applaud the effort that must have gone into collecting all these data, and in general, I am happy with the write up. The finding (that plain trumps stripes) is not a surprise to me, having had serious doubts about the generality of dazzle camouflage effectiveness. I think, with a cleaner write up, and perhaps some rethinking of the presentation/analysis this will make a great contribution to the literature on dazzle camouflage.

My biggest general concerns relate to two things: plots, and the analysis of fitness increases over the experiment. In particular plots 2 and 4 do a poor job of showing the reader enough information to assess the statements in the text. The fit lines bear no relation to the models in question - and the scales exaggerate the effects to some degree (because of log axis and real data are nowhere to be seen in the plots). In all cases plots could be improved by plotting the model fit and model confidence intervals from the actual model rather than ggplot functions. I expand on

my thinking in my specific comments.

My concern about the analysis of fitness increase is that the plots, and (manuscript wide) concentration on p values rather than effect sizes is a little bit problematic. Looking at this effect in the raw data (graciously provided by authors) it looks as though in the best case the mean fitness (reaction time) might have increased by <200ms. Now, that is not the end of the world, and apparently this was enough of a fitness difference to drive selection in the evolutionary algorithm, but I think it certainly is important to the readers' understanding of the study. It's really quite a small difference compared to the scale of the noise in the data – and I feel the reader ought to know that (and not have to interpret a log scale smoothed gam line with no data in it to do so).

Another general concern is that the paper contains many models of various sorts, but nowhere can we explicitly see what the model was, the degrees of freedom, the effect sizes/coefficients etc. I think a series of tables indicating the models used would be appropriate.

Specific comments:

50: Citations in plain text and not numbered.

115: Could targets have evolved in speed too? This approach could ratify the results in the context of broader theory in that you might expect that targets that stay still evolve to match the background, whereas targets that move would (presumably given your results) lose texture information. Or just add a population of targets that never move. Not a suggestion for the current paper but thoughts for possible future investigation.

88: While you couldn't get participant demographics, it might be nice to know whether those demographics changed on large scales. For instance, did the game run on weekdays as well as weekends, and school holidays; would the expected visitors change? Would they change over the day? Could these factors account for the apparent auto-correlated peaks and troughs in the fitness data? I realize it's probably too late for that here, but might be a factor for future experiments.

113: Please clarify whether the angles actually random but less than 90 or always 90 degrees from previous? If they are random is this drawn from a uniform distribution (jerky movement) or a gaussian distribution (smoother motion, if so what SD?). If it was always 90 degrees, isn't that quite weird movement?

118: Regarding the time-out, some lines in the provided data have fitness larger than 5000 (ms), and a number of values are negative, I assume these are errors. Actually, in replicate one there is also a pretty dense cluster of points all at 5000ms in the final generation – are these errors? Are they omitted? Especially problematic given they are in the same direction of the hypothesis – although vastly outweighed by the remaining data. Equally, the last generation of the null model appear to have much greater variation than the rest of the trials (all the points below ~ -0.5 on $\log(\text{fitness})$ scale measured in seconds). I would consider carefully checking the code that generated the null model to work out why. I really appreciate the author's sharing the data, but it might be worthwhile adding to the readme more specific data regarding the columns in the data ('GaborMin_a3_s4', and 'SD' aren't particularly understandable).

121: Caption overlaps figure.

129: I think that the generation of patterns could be explained to a greater degree in this section. It's not too clear that all patterns are complex manipulations of a set of starting images. It's also not clear if this method of generating patterns was developed de-novo, or based on other approaches/is conventional. Would it have been possible to generate patterns with reaction

diffusion algorithms or similar?

157: I very much like that you ran the control model as it helps to control for drift and other complex processes that could be going on with the experiment!

162: Does this approach really mesh with your modelling approach which assumes that fitness is lognormally distributed? A histogram of the fitness from these null experiments certainly looks pretty perfectly normal. I wonder if this has implications for the comparisons.

167: 'Pattern quantification and statistical analysis'? I'd actually consider separating these parts into two sections. That would also help the reader because later where you discuss the models selection and then regression of the five final metrics you could refer them back to this section.

186: Change 'measures of patterning' to 'pattern measures' then use consistently throughout? Elsewhere this appears as camouflage metrics.

187: I feel that this explanation needs expansion, I think it's more true to say that you reduced variables through a model selection approach (since all of the modelling is done with glms). Even that doesn't really make clear what you did (you could have used forward or backward term selection on all effects). Reading the SI, I think that you individually made models for each of an array of pattern variables in sets of categories, and picked the lowest AIC value model in each case? I think overall the MS is pretty light on details on the statistical approach, to the point that it's pretty opaque to the reader. I would also list the pattern metrics that you end up with, or at least mention that there were five since you (193, 198) mention five 'camouflage metrics', but it's not clear that these come from this analysis.

189: This sentence could probably more accurately read: "We generated a linear model of fitness against generation (with replicate, and target speed as fixed effects) to test whether there was a change in fitness across generations." This will help the reader to understand the methods here. The same kinds of changes will help elsewhere.

190: Equally, this could be clearer if you spell out what you did more explicitly. I also don't really understand how you dealt with replicate number in this particular model - you gave the control experiment an arbitrary replicate number and still included replicate as a fixed factor? I feel that you might be better off independently generating estimates for the slope for real and control experiments and then (qualitatively) comparing them. By my estimates in fast experiments there's something like a ~200 ms increase in fitness over (100) generations - and basically no improvement in the control experiments. I think that might be enough to indicate the difference, rather than a significant interaction term.

194: I assume the first 40 generations were used because all replicates have at least 40, but you should make that clear for the reader here. Are all analyses done with just the 40 generations? Or do you mix and match? If all analyses were on 40 generations, perhaps plots should reflect that. You could then put (un-aggregated) plots of the whole experiments in SI?

195: I'd personally lose the 'we wanted to' for slightly more formal phrasing.

193 + 198: Are these the same 5 pattern metrics; not clear where 'camouflage metrics' comes from.

203: 'Motion modelling methods'? All analyses used models.

205: Citation in plain text, not sure if this was deliberate in this case though.

Figure 2: I dislike this graph for a number of reasons, and think the authors should carefully consider changing it. In general it is good practice to show the actual data, I also feel that there is no need for a summary plot like this to be on a log(y) scale. On top of that the smoothed GAMs

bear no relation to the actual analyses/models run on these data – so I don't really know why it's there on its own (rather than potentially as an extra). It would also be nice for the authors to add their carefully generated null model to this so the reader can eyeball the differences. Attached is a version of something that I think would work better, panel columns being population, rows being replicate. Then overlaid are estimated linear regression lines (from a simple linear Bayesian modelling approach, just for an example). I would say that at the least the lines and CI's should represent predictions from the model actually used (which can relatively easily be converted back from log scale so the whole thing can be in natural units).

229: I feel you should make it clear, perhaps somewhere in the methods, that this is always the natural logarithm. In fact, using the log of fitness to make it normal is only mentioned in the SI as far as I can tell. I think that should also occur in the methods. Plotting histograms of fitness, it also appears that the log-normality is only really clear across the board when you aggregate across conditions. Fitness looks pretty normal for fast targets across the board (also see another comment on the null model approach – all these fitness-es look normal not lognormal). Then slower conditions are lognormal, or potentially bi-modally distributed in slow conditions. I don't know if you need to consider this in your chosen statistical approach.

237: Looking at the data I wonder where the censored data are: I.e. failures to catch a target before timeout. Would looking at those data be interesting? Hard to say, I suppose failure to catch could be just not finishing the experiment rather than actually failing for the whole time. Might be nice to note.

241: Why did you choose to model generation as a second order polynomial? I don't think this is addressed anywhere in the paper/SI. Plotting $\log(\text{Fitness}) \sim \text{generation}$ (dis-aggregated, and even just first trials) doesn't indicate to me that the relationship is nonlinear. I would like some explanation (or perhaps a comparison of nested models) of why this approach was taken. The following sentence points to the GAM figure (comments above for why I think that is a bit problematic), and to figure s5 – where the leveling off may only be apparent because of a ggplot spline fit, which I would also take with a bit of salt (not representing any used model fit, and plotted with dis-aggregated data). One argument may be that you would expect fitness to drop off as you got to better values, as targets converge on the 'best' texture. However, if the data don't show that (and it's not too clear to me that they do), the poly seems unnecessary or might even be inappropriate.

258: I like this plot! It might be nice (in the SI) to have more of these kinds of plots so an interested reader can really look at the kinds of textures tested (they look great!).

Figure 4: Again, these GAM (?) fits don't really bear relation to the models you are talking about. My thoughts are the same here as above. I would like to see real data, and a predicted fit line etc, so that I can judge both the model and the interpretation. The figure is also clipped on the right. It may be that a (better) summary figure here with dis-aggregated data would work, if you include in the SI plots of the real data and model fits.

Figure 4 caption: Change in -some- parameters, those selected by a process looking for values that change differently between the experimental and control conditions. Maybe clarify for the reader.

273: Since the result is that targets lose pattern overall, I wonder if you could take an alternative approach to these analyses. Rather than using model selection to find the variables that best predict fitness could you generate a lower dimensional space that controls for correlations between variables (i.e. PCA). If you did PCA on your measured pattern variables, you would expect that some locus would represent the un-patterned targets, surrounded by the more variable targets. If you then overlaid fitness outcomes on the PCA plot (either with color or z height) you'd also expect that the un-patterned locus would be an island of higher fitness right? Given the volume of data, you might need kernel density estimates or similar to summarize

fitness reasonably. I think this would make the same point – that across pattern variables, un-patterned targets had higher fitness, but it obviates the need to throw away the other 60 odd variables you measured. A similar plot using just the null data should just show noise. What you'd end up with is something like a fitness landscape in your lower dimensional pattern space. This may even fit in with your later analysis of selection rates if the PCA components are somewhat interpretable.

277: Camouflage metrics or pattern metrics?

Figure 5: I like seeing data in the plots!

283: When you say differences in selection rates across generation, do you mean that the polynomial term/s for generation were significant? Or do you mean differences between populations of fast medium and slow?

299: Why the most coherent targets? You indicate in the SI that it makes no qualitative difference, but the choice isn't qualified anywhere I can see. Again on 318.

Figure 6 & 7, elsewhere fitness plotted in log(seconds), SI indicates that you did ran models on log fitness here too, so the linear model here doesn't really represent the models. I actually prefer plots in natural units like this though, with transformed fit lines from the actual models.

334: 'As this was correlated with capture time' perhaps.

340: This is a fantastic point - pretty much wherever people use plain targets they seem better or equivalent.

365: Perhaps because the task as devised also includes an element of detection? I.e. to capture a target the participant must detect it, predict the trajectory, and accurately move their hand to capture it. It's conceivable that the fitness changes seen across the experiment are caused by delays in detection rather than tracking/capture per se. You might argue that your experiment better reflects a real predation event, but is less representative of specific hypotheses about target tracking.

372: Interesting point, though I suppose mutations affecting the rotation of stripes in real animals are not small? I.e. I don't think it's as easy for a tiger to rotate its stripes as it is for your targets.

375: Might be a relevant place to cite some literature on distance dependence in stripes for camouflage/aposematism here. And/or some literature on stripes for communication.

389: Very nice paragraph. As an aside, I think you could modify this experiment in many interesting ways. For instance, even just inverting the experiment so it has selection for the easiest targets to catch – would this converge on a dazzle pattern? Equally, if bugs could evolve in a few movement parameters (speed and predictability perhaps) I wonder if you would end up with different populations taking different strategies.

404: And animals are unlikely to always have a birds-eye view of a single bug moving around in an enclosed box for 5 seconds! (Again, totally in line with the literature but it does jump out as equally important to the specifics of the visual system).

Supplemental information

So in figure S1 the three central images the final result or just the very center one? I find the explanation a little hard to follow in the context of the figure. A big montage of simulated images might be nice to show the range of the patterns possible. Equally, what inspired this approach

and is it conventional?

The last line of mutation process indicates that even for the half of bugs that are copied with mutation into the next generation, most were not mutated – is that right? It might make sense to make that clear in the main text?

In S3, it might be nice to have a table of the models tested so that a reader might know how many variables you sifted through to find ones predictive of fitness. I do wonder if you generated an equal number of randomly generated null pattern metrics how many you would end up with significantly predicting fitness/low AIC. Might also be nice to know how you did these analyses (command/framework for comparisons in r?). How did you end up with multiple best models in the luminance category, were they all below delta X AIC (if so what was X?) or are there multiple categories in this set? Basically, I think more explanation is needed.

S5: I feel that dis-aggregated plots (replicate and population) would be more informative to the reader. It might also be a nice place to show the data in natural units (s or ms) too.

Decision letter (RSPB-2020-0899.R0)

29-May-2020

Dear Dr Hughes:

I am writing to inform you that your manuscript RSPB-2020-0899 entitled "The evolution of patterning during movement in a large-scale citizen science game" has, in its current form, been rejected for publication in Proceedings B.

This action has been taken on the advice of referees, who have recommended that substantial revisions are necessary. With this in mind we would be happy to consider a resubmission, provided the comments of the referees are fully addressed. However please note that this is not a provisional acceptance.

Sincerely,
Dr Sasha Dall
mailto:proceedingsb@royalsociety.org

Associate Editor

Board Member: 1

Comments to Author:

This paper uses a citizen science approach to examine selection on the colour pattern of animated prey in an evolutionary game. This approach (human-imposed selection to examine evolution of traits in an artificial environment) has been used successfully with much smaller sample sizes of subjects and here it is being used to tackle an interesting biological question about the function and evolution of motion dazzle illusions. I share the reviewers' enthusiasm for the study but as you can see, both raise some concerns which need to be addressed. In particular, I'd like to emphasise the need to provide thorough documentation for the open source code provided with the study. This is important to ensure the utility of the study and its approach to the scientific community. Also, there is an opportunity to further explore some of the mechanisms relating movement to pattern – reviewer 1 has some good suggestions here. Reviewer 2 raises some important concerns regarding analyses and presentation of the data. The model structure, outputs and especially the effect sizes need to be clearly shown for the reader to evaluate the results. There may be significant effects, especially given the large sample size, but is the effect size sufficient to be biologically meaningful? Carefully addressing reviewer 2's detailed comments will greatly improve the manuscript.

Reviewer(s)' Comments to Author:

Referee: 1

Comments to the Author(s)

This is a well-written, concise article with a very precisely documented methodology. I find the citizen science approach for data collection a versatile method and providing an off-the-shelf touchscreen software for other behavioural ecologists to run camouflage experiments would be of great interest. However, I have a few concerns regarding the paper which are listed below:

1. The authors should provide a readme file on the github repository that provides the necessary information on how to run the software locally. Right now, the readme only says "A new camouflage hunting evolution experiment where we incorporate movement. More info soon!". If the authors only wish to update this information upon publication (makes sense to me), it would be still useful to make the readme available to reviewers.

2. Between lines 369-377 the authors argue that the orientation of stripes could have a significant effect on fitness and small mutations can render stripes a relatively unstable strategy. I wonder whether the random movement of targets could play a significant role in the instability of stripes and their lower fitness compared to uniform grey. Classic examples of stripy targets (where stripes thought to give protection from attacks) like zebras and ships cannot change course so rapidly as the targets in this study and the orientation of their stripes remain consistent to their trajectory, i.e. neither zebras or ships can "fly" upwards/downwards from the predators point of view. It would be interesting to see if a more consistent trajectory (in terms of less angular changes) yields better fitness for stripy patterns. I am not necessarily suggesting further experiments, but I think at least the difference between randomly moving targets and beasts bounded by the laws of physics should be expanded more in the discussion. I guess a real-world example of a stripy animal that can very quickly change its trajectory would be a hoverfly, but as far as I remember there is an argument that they mimick bees...

3. I wonder if there is a correlation between the number of capture attempts / location of capture attempts and the patterns. Could the edges of relatively fast-moving uniform grey targets blur

with the backgrounds while the edges of high contrast targets remain easily trackable? I remember that classic high contrast dazzle patterns should make the predator under or overshoot the target along its expected trajectory, however this effect could be significantly lowered by targets randomly changing trajectory while the “blurring” mentioned above might be independent from how fast directional changes occur (if they occur at all).

Referee: 2

Comments to the Author(s)

This is an ambitious and well designed experiment that uses an evolutionary algorithm to find the best pattern for escaping capture in a human-predator style experiment. I applaud the effort that must have gone into collecting all these data, and in general, I am happy with the write up. The finding (that plain trumps stripes) is not a surprise to me, having had serious doubts about the generality of dazzle camouflage effectiveness. I think, with a cleaner write up, and perhaps some rethinking of the presentation/analysis this will make a great contribution to the literature on dazzle camouflage.

My biggest general concerns relate to two things: plots, and the analysis of fitness increases over the experiment. In particular plots 2 and 4 do a poor job of showing the reader enough information to assess the statements in the text. The fit lines bear no relation to the models in question – and the scales exaggerate the effects to some degree (because of log axis and real data are nowhere to be seen in the plots). In all cases plots could be improved by plotting the model fit and model confidence intervals from the actual model rather than ggplot functions. I expand on my thinking in my specific comments.

My concern about the analysis of fitness increase is that the plots, and (manuscript wide) concentration on p values rather than effect sizes is a little bit problematic. Looking at this effect in the raw data (graciously provided by authors) it looks as though in the best case the mean fitness (reaction time) might have increased by <200ms. Now, that is not the end of the world, and apparently this was enough of a fitness difference to drive selection in the evolutionary algorithm, but I think it certainly is important to the readers’ understanding of the study. It’s really quite a small difference compared to the scale of the noise in the data – and I feel the reader ought to know that (and not have to interpret a log scale smoothed gam line with no data in it to do so).

Another general concern is that the paper contains many models of various sorts, but nowhere can we explicitly see what the model was, the degrees of freedom, the effect sizes/coefficients etc. I think a series of tables indicating the models used would be appropriate.

Specific comments:

50: Citations in plain text and not numbered.

115: Could targets have evolved in speed too? This approach could ratify the results in the context of broader theory in that you might expect that targets that stay still evolve to match the background, whereas targets that move would (presumably given your results) lose texture information. Or just add a population of targets that never move. Not a suggestion for the current paper but thoughts for possible future investigation.

88: While you couldn’t get participant demographics, it might be nice to know whether those demographics changed on large scales. For instance, did the game run on weekdays as well as weekends, and school holidays; would the expected visitors change? Would they change over the day? Could these factors account for the apparent auto-correlated peaks and troughs in the fitness data? I realize it’s probably too late for that here, but might be a factor for future experiments.

113: Please clarify whether the angles actually random but less than 90 or always 90 degrees from previous? If they are random is this drawn from a uniform distribution (jerky movement) or a gaussian distribution (smoother motion, if so what SD?). If it was always 90 degrees, isn't that quite weird movement?

118: Regarding the time-out, some lines in the provided data have fitness larger than 5000 (ms), and a number of values are negative, I assume these are errors. Actually, in replicate one there is also a pretty dense cluster of points all at 5000ms in the final generation – are these errors? Are they omitted? Especially problematic given they are in the same direction of the hypothesis – although vastly outweighed by the remaining data. Equally, the last generation of the null model appear to have much greater variation than the rest of the trials (all the points below ~ -0.5 on $\log(\text{fitness})$ scale measured in seconds). I would consider carefully checking the code that generated the null model to work out why. I really appreciate the author's sharing the data, but it might be worthwhile adding to the readme more specific data regarding the columns in the data ('GaborMin_a3_s4', and 'SD' aren't particularly understandable).

121: Caption overlaps figure.

129: I think that the generation of patterns could be explained to a greater degree in this section. It's not too clear that all patterns are complex manipulations of a set of starting images. It's also not clear if this method of generating patterns was developed de-novo, or based on other approaches/is conventional. Would it have been possible to generate patterns with reaction diffusion algorithms or similar?

157: I very much like that you ran the control model as it helps to control for drift and other complex processes that could be going on with the experiment!

162: Does this approach really mesh with your modelling approach which assumes that fitness is lognormally distributed? A histogram of the fitness from these null experiments certainly looks pretty perfectly normal. I wonder if this has implications for the comparisons.

167: 'Pattern quantification and statistical analysis'? I'd actually consider separating these parts into two sections. That would also help the reader because later where you discuss the models selection and then regression of the five final metrics you could refer them back to this section.

186: Change 'measures of patterning' to 'pattern measures' then use consistently throughout? Elsewhere this appears as camouflage metrics.

187: I feel that this explanation needs expansion, I think it's more true to say that you reduced variables through a model selection approach (since all of the modelling is done with glms). Even that doesn't really make clear what you did (you could have used forward or backward term selection on all effects). Reading the SI, I think that you individually made models for each of an array of pattern variables in sets of categories, and picked the lowest AIC value model in each case? I think overall the MS is pretty light on details on the statistical approach, to the point that it's pretty opaque to the reader. I would also list the pattern metrics that you end up with, or at least mention that there were five since you (193, 198) mention five 'camouflage metrics', but it's not clear that these come from this analysis.

189: This sentence could probably more accurately read: "We generated a linear model of fitness against generation (with replicate, and target speed as fixed effects) to test whether there was a change in fitness across generations." This will help the reader to understand the methods here. The same kinds of changes will help elsewhere.

190: Equally, this could be clearer if you spell out what you did more explicitly. I also don't really understand how you dealt with replicate number in this particular model – you gave the control

experiment an arbitrary replicate number and still included replicate as a fixed factor? I feel that you might be better off independently generating estimates for the slope for real and control experiments and then (qualitatively) comparing them. By my estimates in fast experiments there's something like a ~200 ms increase in fitness over (100) generations – and basically no improvement in the control experiments. I think that might be enough to indicate the difference, rather than a significant interaction term.

194: I assume the first 40 generations were used because all replicates have at least 40, but you should make that clear for the reader here. Are all analyses done with just the 40 generations? Or do you mix and match? If all analyses were on 40 generations, perhaps plots should reflect that. You could then put (un-aggregated) plots of the whole experiments in SI?

195: I'd personally lose the 'we wanted to' for slightly more formal phrasing.

193 + 198: Are these the same 5 pattern metrics; not clear where 'camouflage metrics' comes from.

203: 'Motion modelling methods'? All analyses used models.

205: Citation in plain text, not sure if this was deliberate in this case though.

Figure 2: I dislike this graph for a number of reasons, and think the authors should carefully consider changing it. In general it is good practice to show the actual data, I also feel that there is no need for a summary plot like this to be on a log(y) scale. On top of that the smoothed GAMs bear no relation to the actual analyses/models run on these data – so I don't really know why it's there on its own (rather than potentially as an extra). It would also be nice for the authors to add their carefully generated null model to this so the reader can eyeball the differences. Attached is a version of something that I think would work better, panel columns being population, rows being replicate. Then overlaid are estimated linear regression lines (from a simple linear Bayesian modelling approach, just for an example). I would say that at the least the lines and CI's should represent predictions from the model actually used (which can relatively easily be converted back from log scale so the whole thing can be in natural units).

229: I feel you should make it clear, perhaps somewhere in the methods, that this is always the natural logarithm. In fact, using the log of fitness to make it normal is only mentioned in the SI as far as I can tell. I think that should also occur in the methods. Plotting histograms of fitness, it also appears that the log-normality is only really clear across the board when you aggregate across conditions. Fitness looks pretty normal for fast targets across the board (also see another comment on the null model approach – all these fitness-es look normal not lognormal). Then slower conditions are lognormal, or potentially bi-modally distributed in slow conditions. I don't know if you need to consider this in your chosen statistical approach.

237: Looking at the data I wonder where the censored data are: I.e. failures to catch a target before timeout. Would looking at those data be interesting? Hard to say, I suppose failure to catch could be just not finishing the experiment rather than actually failing for the whole time. Might be nice to note.

241: Why did you choose to model generation as a second order polynomial? I don't think this is addressed anywhere in the paper/SI. Plotting log(Fitness) ~ generation (dis-aggregated, and even just first trials) doesn't indicate to me that the relationship is nonlinear. I would like some explanation (or perhaps a comparison of nested models) of why this approach was taken. The following sentence points to the GAM figure (comments above for why I think that is a bit problematic), and to figure s5 – where the leveling off may only be apparent because of a ggplot spline fit, which I would also take with a bit of salt (not representing any used model fit, and plotted with dis-aggregated data). One argument may be that you would expect fitness to drop off as you got to better values, as targets converge on the 'best' texture. However, if the data don't

show that (and it's not too clear to me that they do), the poly seems unnecessary or might even be inappropriate.

258: I like this plot! It might be nice (in the SI) to have more of these kinds of plots so an interested reader can really look at the kinds of textures tested (they look great!).

Figure 4: Again, these GAM (?) fits don't really bear relation to the models you are talking about. My thoughts are the same here as above. I would like to see real data, and a predicted fit line etc, so that I can judge both the model and the interpretation. The figure is also clipped on the right. It may be that a (better) summary figure here with dis-aggregated data would work, if you include in the SI plots of the real data and model fits.

Figure 4 caption: Change in -some- parameters, those selected by a process looking for values that change differently between the experimental and control conditions. Maybe clarify for the reader.

273: Since the result is that targets lose pattern overall, I wonder if you could take an alternative approach to these analyses. Rather than using model selection to find the variables that best predict fitness could you generate a lower dimensional space that controls for correlations between variables (i.e. PCA). If you did PCA on your measured pattern variables, you would expect that some locus would represent the un-patterned targets, surrounded by the more variable targets. If you then overlaid fitness outcomes on the PCA plot (either with color or z height) you'd also expect that the un-patterned locus would be an island of higher fitness right? Given the volume of data, you might need kernel density estimates or similar to summarize fitness reasonably. I think this would make the same point - that across pattern variables, un-patterned targets had higher fitness, but it obviates the need to throw away the other 60 odd variables you measured. A similar plot using just the null data should just show noise. What you'd end up with is something like a fitness landscape in your lower dimensional pattern space. This may even fit in with your later analysis of selection rates if the PCA components are somewhat interpretable.

277: Camouflage metrics or pattern metrics?

Figure 5: I like seeing data in the plots!

283: When you say differences in selection rates across generation, do you mean that the polynomial term/s for generation were significant? Or do you mean differences between populations of fast medium and slow?

299: Why the most coherent targets? You indicate in the SI that it makes no qualitative difference, but the choice isn't qualified anywhere I can see. Again on 318.

Figure 6 & 7, elsewhere fitness plotted in log(seconds), SI indicates that you did ran models on log fitness here too, so the linear model here doesn't really represent the models. I actually prefer plots in natural units like this though, with transformed fit lines from the actual models.

334: 'As this was correlated with capture time' perhaps.

340: This is a fantastic point - pretty much wherever people use plain targets they seem better or equivalent.

365: Perhaps because the task as devised also includes an element of detection? I.e. to capture a target the participant must detect it, predict the trajectory, and accurately move their hand to capture it. It's conceivable that the fitness changes seen across the experiment are caused by delays in detection rather than tracking/capture per se. You might argue that your experiment

better reflects a real predation event, but is less representative of specific hypotheses about target tracking.

372: Interesting point, though I suppose mutations affecting the rotation of stripes in real animals are not small? I.e. I don't think it's as easy for a tiger to rotate its stripes as it is for your targets.

375: Might be a relevant place to cite some literature on distance dependence in stripes for camouflage/aposematism here. And/or some literature on stripes for communication.

389: Very nice paragraph. As an aside, I think you could modify this experiment in many interesting ways. For instance, even just inverting the experiment so it has selection for the easiest targets to catch – would this converge on a dazzle pattern? Equally, if bugs could evolve in a few movement parameters (speed and predictability perhaps) I wonder if you would end up with different populations taking different strategies.

404: And animals are unlikely to always have a birds-eye view of a single bug moving around in an enclosed box for 5 seconds! (Again, totally in line with the literature but it does jump out as equally important to the specifics of the visual system).

Supplemental information

So in figure S1 the three central images the final result or just the very center one? I find the explanation a little hard to follow in the context of the figure. A big montage of simulated images might be nice to show the range of the patterns possible. Equally, what inspired this approach and is it conventional?

The last line of mutation process indicates that even for the half of bugs that are copied with mutation into the next generation, most were not mutated – is that right? It might make sense to make that clear in the main text?

In S3, it might be nice to have a table of the models tested so that a reader might know how many variables you sifted through to find ones predictive of fitness. I do wonder if you generated an equal number of randomly generated null pattern metrics how many you would end up with significantly predicting fitness/low AIC. Might also be nice to know how you did these analyses (command/framework for comparisons in r ?). How did you end up with multiple best models in the luminance category, were they all below ΔX AIC (if so what was X ?) or are there multiple categories in this set? Basically, I think more explanation is needed.

S5: I feel that dis-aggregated plots (replicate and population) would be more informative to the reader. It might also be a nice place to show the data in natural units (s or ms) too.

Author's Response to Decision Letter for (RSPB-2020-0899.R0)

See Appendix A.

RSPB-2020-2823.R0

Review form: Reviewer 1 (Laszlo Talas)

Recommendation

Accept with minor revision (please list in comments)

Scientific importance: Is the manuscript an original and important contribution to its field?
Good

General interest: Is the paper of sufficient general interest?
Excellent

Quality of the paper: Is the overall quality of the paper suitable?
Excellent

Is the length of the paper justified?
Yes

Should the paper be seen by a specialist statistical reviewer?
No

Do you have any concerns about statistical analyses in this paper? If so, please specify them explicitly in your report.
No

It is a condition of publication that authors make their supporting data, code and materials available - either as supplementary material or hosted in an external repository. Please rate, if applicable, the supporting data on the following criteria.

Is it accessible?
No

Is it clear?
No

Is it adequate?
No

Do you have any ethical concerns with this paper?
No

Comments to the Author
Dear Authors

You mention in the reviewer's comments document (reviewer 1, first question) that you have updated the readme file on <https://github.com/nebogeo/dazzlebug>, however I cannot see an update that's less than 2 years old (see attached screenshot). I looked at the OSF repository, but that's all data related. I was hoping to try running the dazzlebug game myself to see if it works for the average user, but without a readme I'm a bit stuck...

Cheers
Reviewer 1

Decision letter (RSPB-2020-2823.R0)

07-Dec-2020

Dear Dr Hughes

I am pleased to inform you that your manuscript RSPB-2020-2823 entitled "The evolution of patterning during movement in a large-scale citizen science game" has been accepted for publication in Proceedings B.

The referee(s) have recommended publication, but also suggest some minor revisions to your manuscript. Therefore, I invite you to respond to the referee(s)' comments and revise your manuscript. Because the schedule for publication is very tight, it is a condition of publication that you submit the revised version of your manuscript within 7 days. If you do not think you will be able to meet this date please let us know.

In order to ensure effective and robust dissemination and appropriate credit to authors the dataset(s) used should be fully cited. To ensure archived data are available to readers, authors

should include a 'data accessibility' section immediately after the acknowledgements section. This should list the database and accession number for all data from the article that has been made publicly available, for instance:

If you wish to submit your data to Dryad (<http://datadryad.org/>) and have not already done so you can submit your data via this link [http://datadryad.org/submit?journalID=RSPB&manu=\(Document not available\)](http://datadryad.org/submit?journalID=RSPB&manu=(Document+not+available)) which will take you to your unique entry in the Dryad repository. If you have already submitted your data to dryad you can make any necessary revisions to your dataset by following the above link. Please see <https://royalsociety.org/journals/ethics-policies/data-sharing-mining/> for more details.

Sincerely,
Dr Sasha Dall
<mailto:proceedingsb@royalsociety.org>

Associate Editor
Board Member
Comments to Author:

Thank-you for carefully and thoroughly addressing the reviewers' comments. I have carefully read through the revised manuscript and I must say, it is beautifully and clearly written! The findings should be of broad interest and the data are not over-interpreted. This will make a fine contribution to Proceedings B.

Just one comment - please make sure the Github links in the manuscript are correct and that the readme file in Github has been updated. Currently the readme file just says 'A new camouflage hunting evolution experiment where we incorporate movement. More info soon!'. This is important to update, or the open source code can't be used (as discovered by reviewer 1, who was keen to test out the game). Also you mention in the response that you provide an R script with analysis workflow for reproducibility but this was not uploaded as part of supplementary material and I couldn't see it on Github.

Reviewer(s)' Comments to Author:

Referee: 1
Comments to the Author(s).
Dear Authors

You mention in the reviewer's comments document (reviewer 1, first question) that you have updated the readme file on <https://github.com/nebogeo/dazzlebug>, however I cannot see an update that's less than 2 years old (see attached screenshot). I looked at the OSF repository, but

that's all data related. I was hoping to try running the dazzlebug game myself to see if it works for the average user, but without a readme I'm a bit stuck...

Cheers

Author's Response to Decision Letter for (RSPB-2020-2823.R0)

See Appendix B.

Decision letter (RSPB-2020-2823.R1)

08-Dec-2020

Dear Dr Hughes

I am pleased to inform you that your manuscript entitled "The evolution of patterning during movement in a large-scale citizen science game" has been accepted for publication in Proceedings B.

Open Access

Paper charges

Sincerely,
Editor, Proceedings B
mailto:proceedingsb@royalsociety.org

Appendix A

We thank the reviewers and the editor for their detailed and helpful comments on the manuscript. We have now substantially revised the manuscript to address their concerns, focusing in particular on making sure the code is well documented and improving the analysis and presentation of the data. Please find below our responses to their points, with their comments in bold font and our responses below in normal font. All line numbers refer to the position of the changes in the 'tracked changes' document.

Associate Editor

Board Member: 1

Comments to Author:

This paper uses a citizen science approach to examine selection on the colour pattern of animated prey in an evolutionary game. This approach (human-imposed selection to examine evolution of traits in an artificial environment) has been used successfully with much smaller sample sizes of subjects and here it is being used to tackle an interesting biological question about the function and evolution of motion dazzle illusions. I share the reviewers' enthusiasm for the study but as you can see, both raise some concerns which need to be addressed. In particular, I'd like to emphasise the need to provide thorough documentation for the open source code provided with the study. This is important to ensure the utility of the study and its approach to the scientific community. Also, there is an opportunity to further explore some of the mechanisms relating movement to pattern – reviewer 1 has some good suggestions here. Reviewer 2 raises some important concerns regarding analyses and presentation of the data. The model structure, outputs and especially the effect sizes need to be clearly shown for the reader to evaluate the results. There may be significant effects, especially given the large sample size, but is the effect size sufficient to be biologically meaningful? Carefully addressing reviewer 2's detailed comments will greatly improve the manuscript.

Thank you very much for your comments. As requested, we have now improved the documentation available for the study, providing more thorough information about the results obtained, and an R script that should allow readers to reproduce the statistics carried out. We have also expanded our discussion of how movement is related to pattern, in reference to Reviewer 1's comments (see below for further details). Finally, we have significantly modified the analyses and presentation of the data in response to Reviewer 2's helpful suggestions. In particular, we have included greater focus on the effect sizes and their biological meaning. Unfortunately, due to the way that variance is partitioned in linear mixed models, we cannot calculate standard effect sizes for individual model terms; however, wherever possible, we report unstandardised effect sizes (in the supplementary material), and also give more concrete examples (e.g. how the average reaction times have changed across generations).

Reviewer(s)' Comments to Author:

Referee: 1

Comments to the Author(s)

This is a well-written, concise article with a very precisely documented methodology. I find the citizen science approach for data collection a versatile method and providing an off-the-shelf touchscreen software for other behavioural ecologists to run camouflage experiments would be of great interest. However, I have a few concerns regarding the paper which are listed below:

1. The authors should provide a readme file on the github repository that provides the necessary information on how to run the software locally. Right now, the readme only says "A new camouflage hunting evolution experiment where we incorporate movement. More info soon!". If the authors only wish to update this information upon publication (makes sense to me), it would be still useful to make the readme available to reviewers.

Thank you for pointing out our oversight here – this has now been updated.

2. Between lines 369-377 the authors argue that the orientation of stripes could have a significant effect on fitness and small mutations can render stripes a relatively unstable strategy. I wonder whether the random movement of targets could play a significant role in the instability of stripes and their lower fitness compared to uniform grey. Classic examples of stripy targets (where stripes thought to give protection from attacks) like zebras and ships cannot change course so rapidly as the targets in this study and the orientation of their stripes remain consistent to their trajectory, i.e. neither zebras or ships can "fly" upwards/downwards from the predators point of view. It would be interesting to see if a more consistent trajectory (in terms of less angular changes) yields better fitness for stripy patterns. I am not necessarily suggesting further experiments, but I think at least the difference between randomly moving targets and beasts bounded by the laws of physics should be expanded more in the discussion. I guess a real-world example of a stripy animal that can very quickly change its trajectory would be a hoverfly, but as far as I remember there is an argument that they mimic bees...

This is a very interesting point, and we totally agree that this would be a useful angle for a follow up study to take. We have included further discussion of this idea (L420-424).

3. I wonder if there is a correlation between the number of capture attempts / location of capture attempts and the patterns. Could the edges of relatively fast-moving uniform grey targets blur with the backgrounds while the edges of high contrast targets remain easily trackable? I remember that classic high contrast dazzle patterns should make the predator under or overshoot the target along its expected trajectory, however this effect could be significantly lowered by targets randomly changing trajectory while the “blurring” mentioned above might be independent from how fast directional changes occur (if they occur at all).

Unfortunately, the way the data was collected (as average RTs across 5 participants for each bug) means that we don't have individual-level data on the number of capture attempts and the location of capture attempts, though the fact that motion energy predicted capture success would seem in line with the idea that the grey targets were more 'blurred' and thus less trackable. Reviewer 2 makes a related point that our uniform grey targets may have been both more difficult to detect and track (whereas perhaps the striped targets only interfere with tracking ability and not detection, given that these high contrast targets should be very visible). We have expanded on these points in the discussion (L412-415).

Referee: 2

Comments to the Author(s)

This is an ambitious and well designed experiment that uses an evolutionary algorithm to find the best pattern for escaping capture in a human-predator style experiment. I applaud the effort that must have gone into collecting all these data, and in general, I am happy with the write up. The finding (that plain trumps stripes) is not a surprise to me, having had serious doubts about the generality of dazzle camouflage effectiveness. I think, with a cleaner write up, and perhaps some rethinking of the presentation/analysis this will make a great contribution to the literature on dazzle camouflage.

My biggest general concerns relate to two things: plots, and the analysis of fitness increases over the experiment. In particular plots 2 and 4 do a poor job of showing the reader enough information to assess the statements in the text. The fit lines bear no relation to the models in question – and the scales exaggerate the effects to some degree (because of log axis and real data are nowhere to be seen in the plots). In all cases plots could be improved by plotting the model fit and model confidence intervals from the actual model rather than ggplot functions. I expand on my thinking in my specific comments.

My concern about the analysis of fitness increase is that the plots, and (manuscript wide) concentration on p values rather than effect sizes is a little bit problematic. Looking at this effect in the raw data (graciously provided by authors) it looks as though in the best case the mean fitness (reaction time) might have increased by <200ms. Now, that is not the end of the world, and apparently this was enough of a fitness difference to drive selection in the evolutionary algorithm, but I think it certainly is important to the readers' understanding of the study. It's really quite a small difference compared to the scale of the noise in the data – and I feel the reader ought to know that (and not have to interpret a log scale smoothed gam line with no data in it to do so).

Another general concern is that the paper contains many models of various sorts, but nowhere can we explicitly see what the model was, the degrees of freedom, the effect sizes/coefficients etc. I think a series of tables indicating the models used would be appropriate.

Specific comments:

50: Citations in plain text and not numbered.

This has now been fixed.

115: Could targets have evolved in speed too? This approach could ratify the results in the context of broader theory in that you might expect that targets that stay still evolve to match the background, whereas targets that move would (presumably given your results) lose texture information. Or just add a population of targets that never move. Not a suggestion for the current paper but thoughts for possible future investigation.

We agree that this is an interesting suggestion for future work and would follow on nicely from work carried out by others using a non-genetic algorithm approach (e.g. Hall et al 2013, Stevens et al 2011). We have added a sentence to the discussion to make this suggestion (L447-449).

88: While you couldn't get participant demographics, it might be nice to know whether those demographics changed on large scales. For instance, did the game run on weekdays as well as weekends, and school holidays; would the expected visitors change? Would they change over the day? Could these factors account for the apparent auto-correlated peaks and troughs in the fitness data? I realize it's probably too late for that here, but might be a factor for future experiments.

The game ran throughout the opening hours of the Eden Project. It is likely that the demographics of visitors did change (for example, there were probably more schoolchildren during weekdays, and more families at weekends). However, we imagine that the reason for the peaks and troughs is that each participant was presented with all speeds, so some generations may have had (by chance) participants who were faster or slower than average, and we would expect this to be seen across all speed populations. However, we have now updated Figure 2 based on Reviewer 2's advice; we believe this gives a better sense of the overall trends within the data (L244).

113: Please clarify whether the angles actually random but less than 90 or always 90 degrees from previous? If they are random is this drawn from a uniform distribution (jerky movement) or a gaussian distribution (smoother motion, if so what SD?). If it was always 90 degrees, isn't that quite weird movement?

Apologies for the unclear phrasing here, this has been reworded: the new angle was randomly chosen to be within 90 degrees of the previous angle (L113).

118: Regarding the time-out, some lines in the provided data have fitness larger than 5000 (ms), and a number of values are negative, I assume these are errors. Actually, in replicate one there is also a pretty dense cluster of points all at 5000ms in the final generation – are these errors? Are they omitted? Especially problematic given they are in the same direction of the hypothesis – although vastly outweighed by the remaining data. Equally, the last generation of the null model appear to have much greater variation than the rest of the trials (all the points below ~ -0.5 on $\log(\text{fitness})$ scale measured in seconds). I would consider carefully checking the code that generated the null model to work out why. I really appreciate the author's sharing the data, but it might be worthwhile adding to the readme more specific data regarding the columns in the data ('GaborMin_a3_s4', and 'SD' aren't particularly understandable).

All negative values were removed before analysis (as suggested, these would have been errors). Similarly, we only used generations that had at least 5 attempts (both for the real data and for the null data); the generations you are referring to did not reach this criterion, so were not used in the analysis. We have now uploaded our analysis script to the OSF, which hopefully makes these 'data cleaning' points clearer, as well as including more information about the data cleaning processes in the main manuscript (L189-191).

There are a small number of bugs with fitness slightly over 5000ms. These are possible due to timing errors due to background processes. High fitness scores likely to be caused by this kind of activity were removed internally (i.e. were not included in the data) with a threshold of 10000ms. However, removing those bugs remaining in the data with an average fitness over 5000ms from the analysis made no difference to the overall conclusions. The code to check this has been included in the analysis script.

The readme has now been updated to give information about the column headings.

121: Caption overlaps figure.

This has been corrected.

129: I think that the generation of patterns could be explained to a greater degree in this section. It's not too clear that all patterns are complex manipulations of a set of starting images. It's also not clear if this method of generating patterns was developed de-novo, or based on other approaches/is conventional. Would it have been possible to generate patterns with reaction diffusion algorithms or similar?

We have added further explanation (L134-135) to try to make the basic principle of the pattern generation clearer. The approach was based on the references included (Reynolds 2011, Sims 1991, Koza 1990) – the idea of 'genetic programming' has a long history in the computer science literature. While reaction diffusion algorithms would have been possible, we used this approach for the reasons outlined in the manuscript – it allowed us to avoid artificial bounds, and also allowed a lot of flexibility in the complexity of targets that could evolve.

157: I very much like that you ran the control model as it helps to control for drift and other complex processes that could be going on with the experiment!

Thank you!

162: Does this approach really mesh with your modelling approach which assumes that fitness is lognormally distributed? A histogram of the fitness from these null experiments certainly looks pretty perfectly normal. I wonder if this has implications for the comparisons.

We have redone the null model using a lognormal distribution as suggested and have updated the manuscript where required (e.g. L164, L259, Figures 3 and 4).

167: 'Pattern quantification and statistical analysis'? I'd actually consider separating these parts into two sections. That would also help the reader because later where you discuss the models selection and then regression of the five final metrics you could refer them back to this section.

We have separated this into two sections, as suggested.

186: Change 'measures of patterning' to 'pattern measures' then use consistently throughout? Elsewhere this appears as camouflage metrics.

Thank you for the suggestion, we have now made this more consistent.

187: I feel that this explanation needs expansion, I think it's more true to say that you reduced variables through a model selection approach (since all of the modelling is done with glms). Even that doesn't really make clear what you did (you could have used forward or backward term selection on all effects). Reading the SI, I think that you individually made models for each of an array of pattern variables in sets of categories, and picked the lowest AIC value model in each case? I think overall the MS is pretty light on details on the statistical approach, to the point that it's pretty opaque to the reader. I would also list the pattern metrics that you end up with, or at least mention that there were five since you (193, 198) mention five 'camouflage metrics', but it's not clear that these come from this analysis.

We have attempted to make the fact that our approach was based on model selection clearer (L194) and have also included the fact that there were five pattern metrics (L194-195). We made the decision to keep the statistical details in the main manuscript fairly light, both due to space restrictions and to make our approach as clear as possible for the general reader. However, we have included more detail in this section where possible (and again, the full analysis script will hopefully make this clearer).

189: This sentence could probably more accurately read: "We generated a linear model of fitness against generation (with replicate, and target speed as fixed effects) to test whether there was a change in fitness across generations." This will help the reader to understand the methods here. The same kinds of changes will help elsewhere.

We have modified the sentence as suggested (L197-199), and have updated the details of the statistical models in the remainder of the section.

190: Equally, this could be clearer if you spell out what you did more explicitly. I also don't really understand how you dealt with replicate number in this particular model – you gave the control experiment an arbitrary replicate number and still included replicate as a fixed factor? I feel that you might be better off independently generating estimates for the slope for real and control experiments and then (qualitatively) comparing them. By my estimates in fast experiments there's something like a ~200 ms increase in fitness over (100) generations – and basically no improvement in the control experiments. I think that might be enough to indicate the difference, rather than a significant interaction term.

Replicate number was included as a random effect; for both the control and experimental data, there were multiple independent 'runs' of the experiment. We feel that it is inappropriate to run linear models, given the fact that nested model comparison suggests that using a polynomial provides a better fit to the experimental data (see below, and supplementary material). However, we have now included predictions from the model to make the size of the fitness increase clearer (L260-262).

194: I assume the first 40 generations were used because all replicates have at least 40, but you should make that clear for the reader here. Are all analyses done with just the 40 generations? Or do you mix and match? If all analyses were on 40 generations, perhaps plots should reflect that. You could then put (un-aggregated) plots of the whole experiments in SI?

Apologies for the lack of clarity here. The first analysis, looking at the change in fitness in the experimental populations, uses all experimental data collected (clarified on L199). Elsewhere, we used just the first 40 generations for all replicates (clarified on L201). This ensures that all experimental replicates are comparable. The plots should reflect the number of generations used in the relevant analysis throughout.

195: I'd personally lose the 'we wanted to' for slightly more formal phrasing.

This has been edited (L210).

193 + 198: Are these the same 5 pattern metrics; not clear where 'camouflage metrics' comes from.

Apologies for the confusion here, these are the same 5 metrics, and we have standardised our phrasing to make this clearer.

203: 'Motion modelling methods'? All analyses used models.

This has been edited (L219).

205: Citation in plain text, not sure if this was deliberate in this case though.

This has been edited (L221).

Figure 2: I dislike this graph for a number of reasons, and think the authors should carefully consider changing it. In general it is good practice to show the actual data, I also feel that there is no need for a summary plot like this to be on a log(y) scale. On top of that the smoothed GAMs bear no relation to the actual analyses/models run on these data – so I don't really know why it's there on its own (rather than potentially as an extra). It would also be nice for the authors to add their carefully generated null model to this so the reader can eyeball the differences. Attached is a version of something that I think would work better, panel columns being population, rows being replicate. Then overlaid are estimated linear regression lines (from a simple linear Bayesian modelling approach, just for an example). I would say that at the least the lines and CI's should represent predictions from the model actually used (which can relatively easily be converted back from log scale so the whole thing can be in natural units).

Thank you for these helpful comments (and for the attached image!) We have reworked Figure 2 to take into account your comments; the raw data is now included, and the summary line now reflects model fit data. We have also put each population in its own panel, as per your suggestion. The null model data is available for comparison with the experimental data in the supplementary material – we made this decision based on space limitations in the main manuscript.

229: I feel you should make it clear, perhaps somewhere in the methods, that this is always the natural logarithm. In fact, using the log of fitness to make it normal is only mentioned in the SI as far as I can tell. I think that should also occur in the methods. Plotting histograms of fitness, it also appears that the log-normality is only really clear across the board when you aggregate across conditions. Fitness looks pretty normal for fast targets across the board (also see another comment on the null model approach – all these fitness-es look normal not lognormal). Then slower conditions are lognormal, or potentially bimodally distributed in slow conditions. I don't know if you need to consider this in your chosen statistical approach.

We have now included an explicit statement to this effect in the methods (L196-197). We based our use of the log normal on residual plot inspection.

237: Looking at the data I wonder where the censored data are: i.e. failures to catch a target before timeout. Would looking at those data be interesting? Hard to say, I suppose failure to catch could be just not finishing the experiment rather than actually failing for the whole time. Might be nice to note.

As the data is aggregated across 5 participants, we unfortunately aren't able to look at censored data in the current set up. We have added a sentence to make this clearer on L146-147.

241: Why did you choose to model generation as a second order polynomial? I don't think this is addressed anywhere in the paper/SI. Plotting log(Fitness) ~ generation (dis-aggregated, and even just first trials) doesn't indicate to me that the relationship is nonlinear. I would like some explanation (or perhaps a comparison of nested models) of why this approach was taken. The following sentence points to the GAM figure (comments above for why I think that is a bit problematic), and to figure s5 – where the leveling off may only be apparent because of a ggplot spline fit, which I would also take with a bit of salt (not representing any used model fit, and plotted with dis-aggregated data). One argument may be that you would expect fitness to drop off as you got to better values, as targets converge on the 'best' texture. However, if the data don't show that (and it's not too clear to me that they do), the poly seems unnecessary or might even be inappropriate.

We have carried out nested model comparisons, which show that the more complex polynomial model is preferred; details are now included in the supplemental material (L81-83).

258: I like this plot! It might be nice (in the SI) to have more of these kinds of plots so an interested reader can really look at the kinds of textures tested (they look great!).

Thank you! We have added further examples to the supplementary material (Figure S3).

Figure 4: Again, these GAM (?) fits don't really bear relation to the models you are talking about. My thoughts are the same here as above. I would like to see real data, and a predicted fit line etc, so that I can judge both the model and the interpretation. The figure is also clipped on the right. It may be that a (better) summary figure here with dis-aggregated data would work, if you include in the SI plots of the real data and model fits.

We have included the raw data and disaggregated the different population conditions so it is hopefully easier to see the differences. The difficulty with the model fits in these graphs is that we fit the models using cumulative link models with generation as the dependent variable (because the non-normality of the pattern measures meant that it was extremely difficult to use these as the dependent variable). However, we feel that displaying the graphs with Generation on the x axis is clearer for the reader. The GAM fit lines therefore provide a visual reference point, but we have tried to make clearer in the Figure legend that these do not reflect the model fitted for data analysis (L287-288).

Figure 4 caption: Change in –some- parameters, those selected by a process looking for values that change differently between the experimental and control conditions. Maybe clarify for the reader.

We have reworded this to attempt to make it clearer (L284).

273: Since the result is that targets lose pattern overall, I wonder if you could take an alternative approach to these analyses. Rather than using model selection to find the variables that best predict fitness could you generate a lower dimensional space that controls for correlations between variables (i.e. PCA). If you did PCA on your measured pattern variables, you would expect that some locus would represent the un-patterned targets, surrounded by the more variable targets. If you then overlaid fitness outcomes on the PCA plot (either with color or z height) you'd also expect that the un-patterned locus would be an island of higher fitness right? Given the volume of data, you might need kernel density estimates or similar to summarize fitness reasonably. I think this would make the same point – that across pattern variables, un-patterned targets had higher fitness, but it obviates the need to throw away the other 60 odd variables you measured. A similar plot using just the null data should just show noise. What you'd end up with is something like a fitness landscape in your lower dimensional pattern space. This may even fit in with your later analysis of selection rates if the PCA components are somewhat interpretable.

Thank you for this suggestion. We have now provided further details of our model selection approach (particularly in the R script detailing the analysis) which will hopefully clarify the method; we think that a PCA approach would give very similar results (given that there is likely to be a high degree of correlation between e.g. the multiple spatial frequencies used for each stripe orientation – therefore selecting one of these measures is likely to be approximately equivalent to using all of them in an optimal way) but could be less straightforward to interpret. Given that we want to be able to look at how different aspects of the patterning evolve, our model selection approach seemed like the best option.

277: Camouflage metrics or pattern metrics?

This has been standardised.

Figure 5: I like seeing data in the plots!

283: When you say differences in selection rates across generation, do you mean that the polynomial term/s for generation were significant? Or do you mean differences between populations of fast medium and slow?

Yes, this is now clarified in L309-310.

299: Why the most coherent targets? You indicate in the SI that it makes no qualitative difference, but the choice isn't qualified anywhere I can see. Again on 318.

Apologies for the confusion here: the analysis is carried out with all bugs and shows a significant interaction between bias and coherence in predicting fitness. (The results of this analysis are unaffected by inclusion of outlier points with circular mean difference of greater than 60 degrees). We have rewritten the results to try to make this clearer (L329-333).

Thus, in the figure, we visualise one part of the interaction where we have a hypothesis: we have displayed the bugs with a high median coherence value because of the prediction made by How and Zanker, that strong

motion illusions are expected if targets are highly coherent and biased. They had no strong predictions for the case where the target had low coherence.

In the case of motion energy, both the model and the graph (Fig 6, bottom) use all bugs.

Figure 6 & 7, elsewhere fitness plotted in log(seconds), SI indicates that you did run models on log fitness here too, so the linear model here doesn't really represent the models. I actually prefer plots in natural units like this though, with transformed fit lines from the actual models.

We have updated the figures so that the fit lines are transformed predictions from the model.

334: 'As this was correlated with capture time' perhaps.

This has now been edited (L378).

340: This is a fantastic point - pretty much wherever people use plain targets they seem better or equivalent.

Thanks, we also think this is an important point that deserves highlighting!

365: Perhaps because the task as devised also includes an element of detection? I.e. to capture a target the participant must detect it, predict the trajectory, and accurately move their hand to capture it. It's conceivable that the fitness changes seen across the experiment are caused by delays in detection rather than tracking/capture per se. You might argue that your experiment better reflects a real predation event, but is less representative of specific hypotheses about target tracking.

We agree that both detection and tracking may be important in our set-up (although motion is normally a very strong cue) and have added some extra discussion of these ideas and how they might relate to uniform/striped patterning on L412-415.

372: Interesting point, though I suppose mutations affecting the rotation of stripes in real animals are not small? I.e. I don't think it's as easy for a tiger to rotate its stripes as it is for your targets.

Reviewer 1 made a similar point, and we have included further discussion of the idea that animals are more restricted in their movements than our targets on L420-424.

375: Might be a relevant place to cite some literature on distance dependence in stripes for camouflage/aposematism here. And/or some literature on stripes for communication.

We have included an extra reference here (L451-452).

389: Very nice paragraph. As an aside, I think you could modify this experiment in many interesting ways. For instance, even just inverting the experiment so it has selection for the easiest targets to catch – would this converge on a dazzle pattern? Equally, if bugs could evolve in a few movement parameters (speed and predictability perhaps) I wonder if you would end up with different populations taking different strategies.

Thank you for these suggestions – we agree that there is a lot of potential for further research using this paradigm & have tried to highlight some of these options throughout the discussion.

404: And animals are unlikely to always have a birds-eye view of a single bug moving around in an enclosed box for 5 seconds! (Again, totally in line with the literature but it does jump out as equally important to the specifics of the visual system).

We agree this is an important consideration and have included this as another possible limitation (L454-455).

Supplemental information

So in figure S1 the three central images the final result or just the very center one? I find the explanation a little hard to follow in the context of the figure. A big montage of simulated images might be nice to show the range of the patterns possible. Equally, what inspired this approach and is it conventional?

The very central image is the final one, and we have now clarified this in the text. As mentioned above, the approach is based on genetic programming algorithms, which have been used in computer science for many years. We have also added a montage of simulated images in Figure S3, as requested.

The last line of mutation process indicates that even for the half of bugs that are copied with mutation

into the next generation, most were not mutated – is that right? It might make sense to make that clear in the main text?

We have clarified that the mutations were relatively infrequent in the main text (L150).

In S3, it might be nice to have a table of the models tested so that a reader might know how many variables you sifted through to find ones predictive of fitness. I do wonder if you generated an equal number of randomly generated null pattern metrics how many you would end up with significantly predicting fitness/low AIC. Might also be nice to know how you did these analyses (command/framework for comparisons in r?). How did you end up with multiple best models in the luminance category, were they all below ΔX AIC (if so what was X?) or are there multiple categories in this set? Basically, I think more explanation is needed.

The full list of models tested is now available in the analysis code. We used only one metric for luminance (the standard deviation of the luminance) but tested multiple other options e.g. the mean luminance, max/min luminance and the contrast. In all cases, we chose the model with the lowest AIC value.

S5: I feel that dis-aggregated plots (replicate and population) would be more informative to the reader. It might also be a nice place to show the data in natural units (s or ms) too.

This graph has been updated as suggested.

Appendix B

Thank-you for carefully and thoroughly addressing the reviewers' comments. I have carefully read through the revised manuscript and I must say, it is beautifully and clearly written! The findings should be of broad interest and the data are not over-interpreted. This will make a fine contribution to Proceedings B.

Just one comment - please make sure the Github links in the manuscript are correct and that the readme file in Github has been updated. Currently the readme file just says 'A new camouflage hunting evolution experiment where we incorporate movement. More info soon!'. This is important to update, or the open source code can't be used (as discovered by reviewer 1, who was keen to test out the game). Also you mention in the response that you provide an R script with analysis workflow for reproducibility but this was not uploaded as part of supplementary material and I couldn't see it on Github.

We apologise for this – this was due to a link not being updated properly. The correct link (<https://github.com/fo-am/dazzlebug/>) is now included in the manuscript: the Github contains all the information about the set-up of the game, including a link to an online version for readers to view themselves. The analysis scripts are available along with the raw data in OSF (<https://osf.io/s5wxy/>) – this should now be publicly available, and the data availability statement has been updated to make it clearer where these scripts can be found.

Reviewer(s)' Comments to Author:

Referee: 1

Comments to the Author(s).

Dear Authors

You mention in the reviewer's comments document (reviewer 1, first question) that you have updated the readme file on <https://github.com/nebogeo/dazzlebug>, however I cannot see an update that's less than 2 years old (see attached screenshot). I looked at the OSF repository, but that's all data related. I was hoping to try running the dazzlebug game myself to see if it works for the average user, but without a readme I'm a bit stuck...

Cheers

Apologies, this was due to an old link and this has now been corrected (see above).